# Deep learning for fast simulation of seismic waves in complex media

Ben Moseley[1], Tarje Nissen-Meyer[2], and Andrew Markham[1]

[1]Department of Computer Science, University of Oxford, UK
[2]Department of Earth Sciences, University of Oxford, UK

**Correspondence:** Ben Moseley (bmoseley@robots.ox.ac.uk)

**Abstract.** The simulation of seismic waves is a core task in many geophysical applications. Numerical methods such as Finite Difference (FD) modelling and Spectral Element Methods (SEM) are the most popular techniques for simulating seismic waves, but disadvantages such as their computational cost prohibit their use for many tasks. In this work we investigate the potential of deep learning for aiding seismic simulation in the Solid Earth sciences. We present two deep neural networks which are able to simulate the seismic response at multiple locations in horizontally layered and faulted 2D acoustic media an order of magnitude faster than traditional finite difference modelling. The first network is able to simulate the seismic response in horizontally layered media and uses a WaveNet network architecture design. The second network is significantly more general than the first and is able to simulate the seismic response in faulted media with arbitrary layers, fault properties and an arbitrary location of the seismic source on the surface of the media, using a conditional autoencoder design. We test the sensitivity of the accuracy of both networks to different network hyperparameters, and show that the WaveNet network can be retrained to carry out fast seismic inversion in the same media. We find that are there are challenges when extending our methods to more complex, elastic and 3D Earth models; for example the accuracy of both networks reduces when they are tested on models outside of their training distribution. We discuss further research directions which could address these challenges and potentially yield useful tools for practical simulation tasks.

## 1 Introduction

Seismic simulations are essential for addressing many outstanding questions in geophysics. In seismic hazards analysis, they are a key tool for quantifying the ground motion of potential earthquakes (Boore, 2003; Cui et al., 2010). In oil and gas prospecting, they allow the seismic response of hydrocarbon reservoirs to be modelled (Chopra and Marfurt, 2007; Lumley, 2001). In geophysical surveying they show how the subsurface is illuminated by different survey designs (Xie et al., 2006). In global geophysics they are used to obtain snapshots of the Earth's interior dynamics by tomography (Hosseini et al., 2019; Bozdağ et al., 2016), to decipher source and path effects from individual seismograms (Krischer et al., 2017) and to model wave effects of complex structures (Thorne et al., 2020; Ni et al., 2002). In seismic inversion they are used to estimate the elastic properties of a medium given its seismic response (Tarantola, 1987; Schuster, 2017) and in Full Waveform Inversion (Fichtner, 2010; Virieux and Operto, 2009), a technique used to image the 3D structure of the subsurface, they are used up to tens of thousands of times to improve on estimates of a medium's elastic properties. In planetary science, seismic simulations play a central role in understanding novel recordings on Mars (Van Driel et al., 2019).

Numerous methods exist for simulating seismic waves, the most popular in fully heterogeneous media being Finite Difference (FD) and Spectral Element Methods (SEM) (Igel, 2017; Moczo et al., 2007; Komatitsch and Tromp, 1999). They are able to capture a large range of physics, including the effects of undulating solid-fluid interfaces (Leng et al., 2019), intrinsic attenuation (van Driel and Nissen-Meyer, 2014a) and anisotropy (van Driel and Nissen-Meyer, 2014b). These methods solve for the propagation of the full seismic wavefield by discretising the elastodynamic equations of motion. For an acoustic heterogeneous medium these are given by the scalar linear equation of motion

$$\rho\nabla\cdot\left(\frac{1}{\rho}\nabla p\right) - \frac{1}{v^2}\frac{\partial^2 p}{\partial t^2} = -\rho\frac{\partial^2 f}{\partial t^2} \ , \tag{1}$$

where $p$ is the acoustic pressure, $f$ is a point source of volume injection (the seismic source), and $v = \sqrt{\kappa/\rho}$ is the velocity of the medium, with $\rho$ the density of the medium and $\kappa$ the adiabatic compression modulus (Long et al., 2013).

Whilst FD and spectral element methods are the primary means of simulation in complex media, a major disadvantage of these methods is their computational cost (Bohlen, 2002; Leng et al., 2016). Typical FD or SEM simulations can involve billions of degrees of freedom, and at each time step the wavefield must be iteratively updated at each 3D grid point. For many practical geophysical applications this is often prohibitively expensive. For example, in global seismology one may be interested in modelling waves up to 1 Hz in frequency to resolve small-scale heterogeneities in the mantle and a single simulation of this type with conventional techniques can cost around 40 million CPU hours (Leng et al., 2019). At crustal scales, industrial seismic imaging requires wave modelling up to tens of Hertz in frequency carried out hundreds of thousands of times for each explosion in a seismic survey, and such requirements can easily fill the largest supercomputers on Earth. Any improvement in efficiency is welcome, not least due to the high financial and environmental costs of high-performance computing.

In some applications, large parts of the Earth model may be relatively smooth or simple. This simplicity can be taken advantage of, for example in the complexity-adapted SEM introduced by Leng et al. (2016), and can deliver a large speed-up compared to standard numerical modelling. Pseudo-analytical methods such as ray tracing and amplitude-versus-offset modelling (Aki and Richards, 1980; Vinje et al., 1993) are another approach which can provide significant speed-ups, albeit being approximate. We note that many applications are constrained and driven by a sparse set of observations on the surface of an Earth model. For these applications we are typically only interested in modelling the seismic response at these points to decipher seismic origin or the 3D structure beneath the surface, yet fully numerical methods still need to iterate the entire wavefield through all points in the model at all points in time. Any shortcut to avoid computing these massive 4D wavefields might lead to drastic efficiency improvements. In short, the points above suggest that alternative and advantageous methods to capture accurate wave physics may be possible for these challenging problems.

The field of machine learning has seen an explosion in growth over the last decade. This has been primarily driven by advancements in deep learning, which has provided more powerful algorithms allowing much more difficult problems to be learned (Goodfellow et al., 2016). This progress has led to a surge in the use of deep learning techniques across many areas of science. In particular, deep neural networks have recently shown promise in their ability to make fast yet sufficiently accurate

predictions of physical phenomena (Guo et al., 2016; Lerer et al., 2016; Paganini et al., 2018). These approaches are able to learn about highly non-linear physics and often offer much faster inference times than traditional simulation.

In this work we ask whether the latest deep learning techniques can aid seismic simulation tasks relevant to the Solid Earth sciences. We investigate the use of deep neural networks and discuss the challenges and opportunities when using them for practical seismic simulation tasks. Our contribution is as follows;

- We present two deep neural networks which are able to simulate seismic waves in 2D acoustic media an order of magnitude faster than FD simulation. The first network uses a WaveNet network architecture (van den Oord et al., 2016) and is able to accurately simulate the pressure response from a fixed point source at multiple locations in a horizontally layered velocity model. The second is significantly more general; it uses a conditional autoencoder network design and is able to simulate the seismic response at multiple locations in faulted media with arbitrary layers, fault properties and an arbitrary location of the source on the surface of the media. In contrast to the classical methods both networks simulate the seismic response in a single inference step, without needing to iteratively model the seismic wavefield through time, resulting in a significant speed-up compared to FD simulation.

- We test the sensitivity of the accuracy of both networks to different network designs, present a loss function with a time-varying gain which improves training convergence and show that fast seismic inversion in horizontal layered media can also be carried out by retraining the WaveNet network.

- We find challenges when extending our methods to more complex, elastic and 3D Earth models and discuss further research directions which could address these challenges and yield useful tools for practical simulation tasks.

In Section 2 we consider the simple case of simulating seismic waves in horizontally layered 2D acoustic Earth models using a WaveNet deep neural network. In Section 3 we move on to the task of simulating more complex faulted Earth models. In Section 4 we discuss the challenges of extending our approaches and future research directions.

## 1.1   Related Work

The use of machine learning and neural networks in geophysics is not new (Van Der Baan and Jutten, 2000). For example, Murat and Rudman (1992) used neural networks to carry out automated first break picking, Dowla et al. (1990) used a neural network to discriminate between earthquakes and nuclear explosions and Poulton et al. (1992) used them for electromagnetic inversion of a conductive target. In seismic inversion, Röth and Tarantola (1994) used a neural network to estimate the velocity of 1D, layered, constant thickness velocity profiles from seismic amplitudes and Nath et al. (1999) used neural networks for cross-well travel-time tomography. However, these early approaches only used shallow network designs with small numbers of free parameters which limits the expressivity of neural networks and the complexity of problems they can learn about (Goodfellow et al., 2016).

The field of machine learning has grown rapidly over the last decade, primarily because of advances in deep learning. The availability of larger datasets, discovery of methods which allow deeper networks to be trained and availability of more

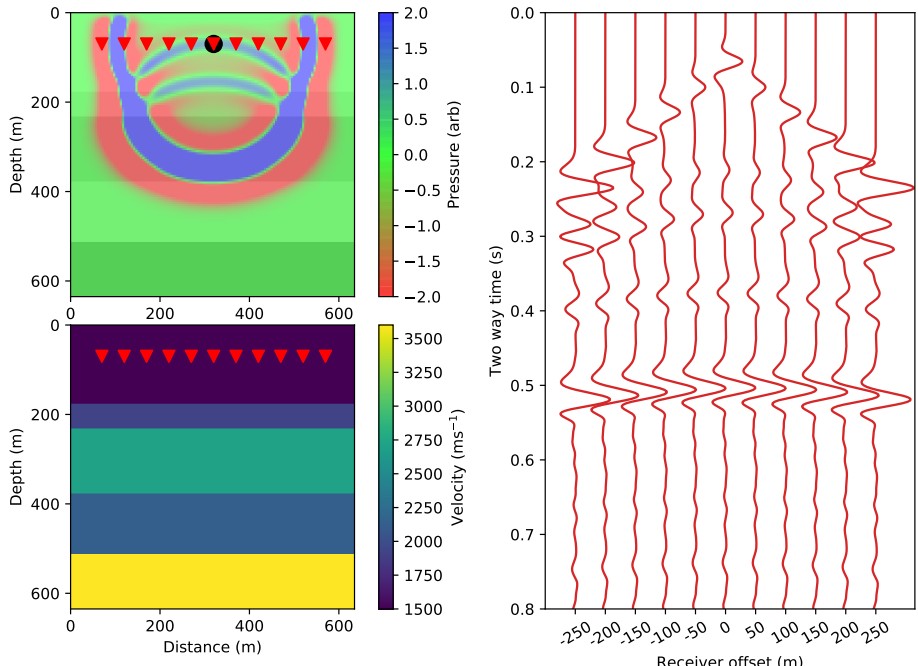

**Figure 1.** Ground truth FD simulation example. Left, top: A 20 Hz Ricker seismic source is emitted close to the surface and propagates through a 2D horizontally layered acoustic Earth model. The black circle shows the source location. 11 receivers are placed at the same depth as the source with a horizontal spacing of 50 m (red triangles). The full wavefield is overlain for a single snapshot in time. Note seismic reflections occur at each velocity interface. Left, bottom: The Earth velocity model. The Earth model has a constant density of 2200 $\mathrm{kgm}^{-2}$. Right: The resulting ground truth pressure response recorded by each of the receivers, using FD modelling. A $t^{2.5}$ gain is applied to the receiver responses for display.

powerful computing architectures (mostly GPUs) has allowed much more complex problems to be learnt (Goodfellow et al., 2016), leading to a surge in the use of deep learning in many different research areas. In physics, Lerer et al. (2016) presented a deep convolutional network which could accurately predict whether randomly stacked wooden towers would fall or remain stable, given 2D images of the tower. Guo et al. (2016) demonstrated that convolutional neural networks could estimate flow

5   fields in complex Computational Fluid Dynamics (CFD) calculations two orders of magnitude faster than a traditional GPU-accelerated CFD solver and Paganini et al. (2018) used a conditional generative adversarial network to simulate particle showers in particle colliders.

A resurgence is occurring in geophysics too (Bergen et al., 2019; Kong et al., 2019). Early examples of deep learning include Devilee et al. (1999), who used deep probabilistic neural networks to estimate crustal thicknesses from surface wave velocities

10   and Valentine and Trampert (2012) who used a deep autoencoder to compress seismic waveforms. More recently, Perol et al. (2018) presented an earthquake identification method using convolutional networks which is orders of magnitude faster than traditional techniques. In seismic inversion, Araya-Polo et al. (2018) proposed an efficient deep learning concept for carrying

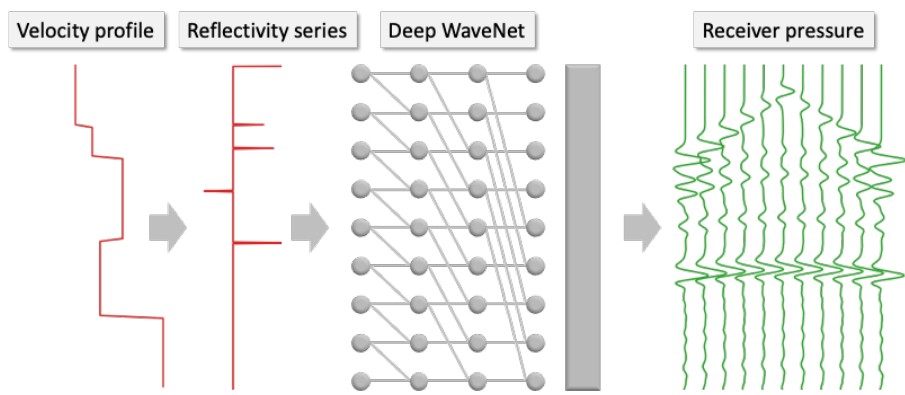

**Figure 2.** Our WaveNet simulation workflow. Given a 1D Earth velocity profile as input (left), our WaveNet deep neural network (middle) outputs a simulation of the pressure responses at the 11 receiver locations in Fig 1. The raw input 1D velocity profile sampled in depth is converted into its normal incidence reflectivity series sampled in time before being input into the network. The network is composed of 9 time-dilated causally-connected convolutional layers with a filter width of 2 and dilation rates which increase exponentially with layer depth. Each hidden layer of the network has same length as the input reflectivity series, 256 channels and a ReLU activation function. A final causally-connected convolutional layer with a filter width of 101 samples, 11 output channels and an identity activation is used to generate the output simulation.

out seismic tomography using the semblance of common mid-point receiver gathers as input. Wu and Lin (2018) proposed a convolutional autoencoder network to carry out seismic inversion, whilst Yang and Ma (2019) adapted a U-net network design for the same purpose. Richardson (2018) demonstrated that a recurrent neural network framework can be used to carry out FWI. Sun and Demanet (2018) showed a method for using deep learning to extrapolate low frequency seismic energy to improve
5    the convergence of FWI algorithms. In seismic simulation Zhu et al. (2017) presented a multi-scale convolutional network for predicting the evolution of the full seismic wavefield in heterogeneous media. Their method was able to approximate wavefield kinematics over multiple time steps, although it suffered from the accumulation of error over time and did not offer a reduction in computational time. Moseley et al. (2018) showed that a convolutional network with a recursive loss function can simulate the full wavefield in horizontally layered acoustic media. Krischer and Fichtner (2017) used a generative adversarial network
10    to simulate seismograms from radially symmetric and smooth Earth models.

In this work we present fast methods for simulating seismic waves in horizontally layered and faulted 2D acoustic media, which offer a significant reduction in computation time compared to Zhu et al. (2017). We also present a fast method for seismic inversion of horizontally layered acoustic media, which is more general than the original approach proposed by Röth and Tarantola (1994) because it is able to invert velocity models with varying numbers of layers and varying layer thicknesses.
15    We restrict ourselves to 2D acoustic media and discuss implications for 3D elastic media below.

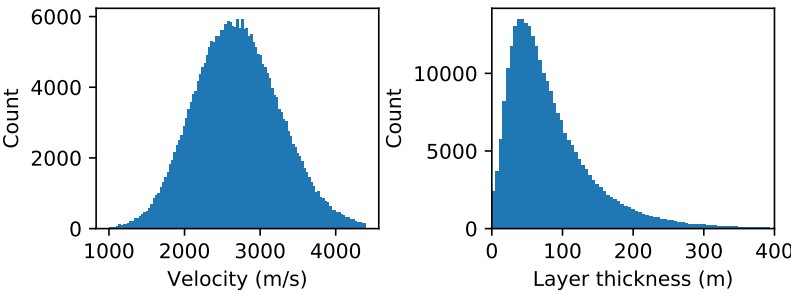

**Figure 3.** Distribution of layer velocity and layer thickness over all examples in the training set.

## 2  Fast seismic simulation in 2D horizontally layered acoustic media using WaveNet

First we consider the simple case of simulating seismic waves in horizontally layered 2D acoustic Earth models. We train a deep neural network with a WaveNet architecture to simulate the seismic response recorded at multiple receiver locations in the Earth model, horizontally offset from a point source emitted at the surface of the model. As mentioned above, many seismic applications are concerned with sparse observations similar to this setup. A key difference of this approach compared to FD and SEM simulation is that the network computes the seismic response at the surface in a single inference step, without needing to iteratively model the seismic wavefield through time, potentially offering a significant speed-up. Whilst we concentrate on simple velocity models here, more complex faulted Earth models are considered in Section 3.

An example simulation we wish to learn is shown in Fig. 1 and our simulation workflow is shown in Fig. 2. The input to the network is a horizontally layered velocity profile and the output of the network is a simulation of the pressure response recorded at each receiver location. We will now discuss deep neural networks, our WaveNet architecture, our simulation workflow and our training methodology in more detail below.

### 2.1  Deep neural networks and the WaveNet network

A neural network is a network of simple computational elements, known as neurons, which perform mathematical operations on multidimensional arrays, or tensors (Goodfellow et al., 2016). The composition of these neurons together defines a mathematical function of the network's input. Each neuron has a set of free parameters, or weights, which are tuned using optimisation, allowing the network's function to be learned, given a set of training data. In deep learning, the neurons are typically arranged in multiple layers, which allows the network to learn highly non-linear functions.

A standard building block in deep learning is the convolutional layer, where all neurons in the layer share the same weight tensor and each neuron has a limited field of view of its input tensor. The output of the layer is achieved by cross correlating the weight tensor with the input tensor. Multiple weight tensors, or filters, can be used to increase the depth of the output tensor. Such designs have achieved state of the art performance across a wide range of machine learning tasks (Gu et al., 2018).

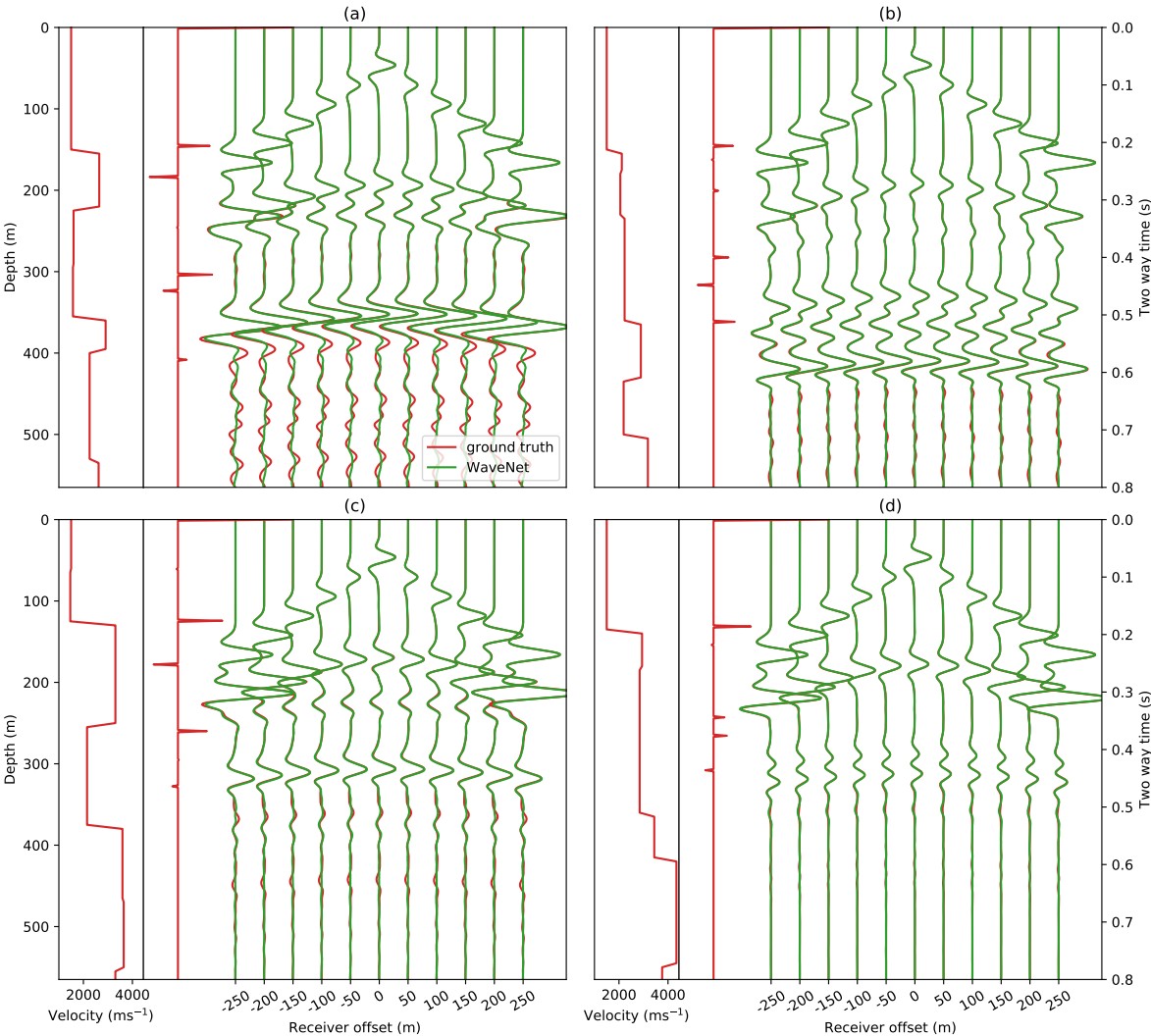

**Figure 4.** WaveNet simulations for 4 randomly selected examples in the test set. Red shows the input velocity model, its corresponding reflectivity series and the ground truth pressure response from FD simulation at the 11 receiver locations. Green shows the WaveNet simulation given the input reflectivity series for each example. A $t^{2.5}$ gain is applied to the receiver responses for display.

The WaveNet network proposed by van den Oord et al. (2016) makes multiple alterations to the standard convolutional layer for its use with time series. Each convolutional layer is made causal; that is, the receptive field of each neuron only contains samples from the input layer whose sample times are before or the same as the current neuron's sample time. Furthermore the WaveNet exponentially dilates the width of its causal connections with layer depth. This allows the field of view of its neurons to increase exponentially with layer depth, without needing a large number of layers. These modifications are made to

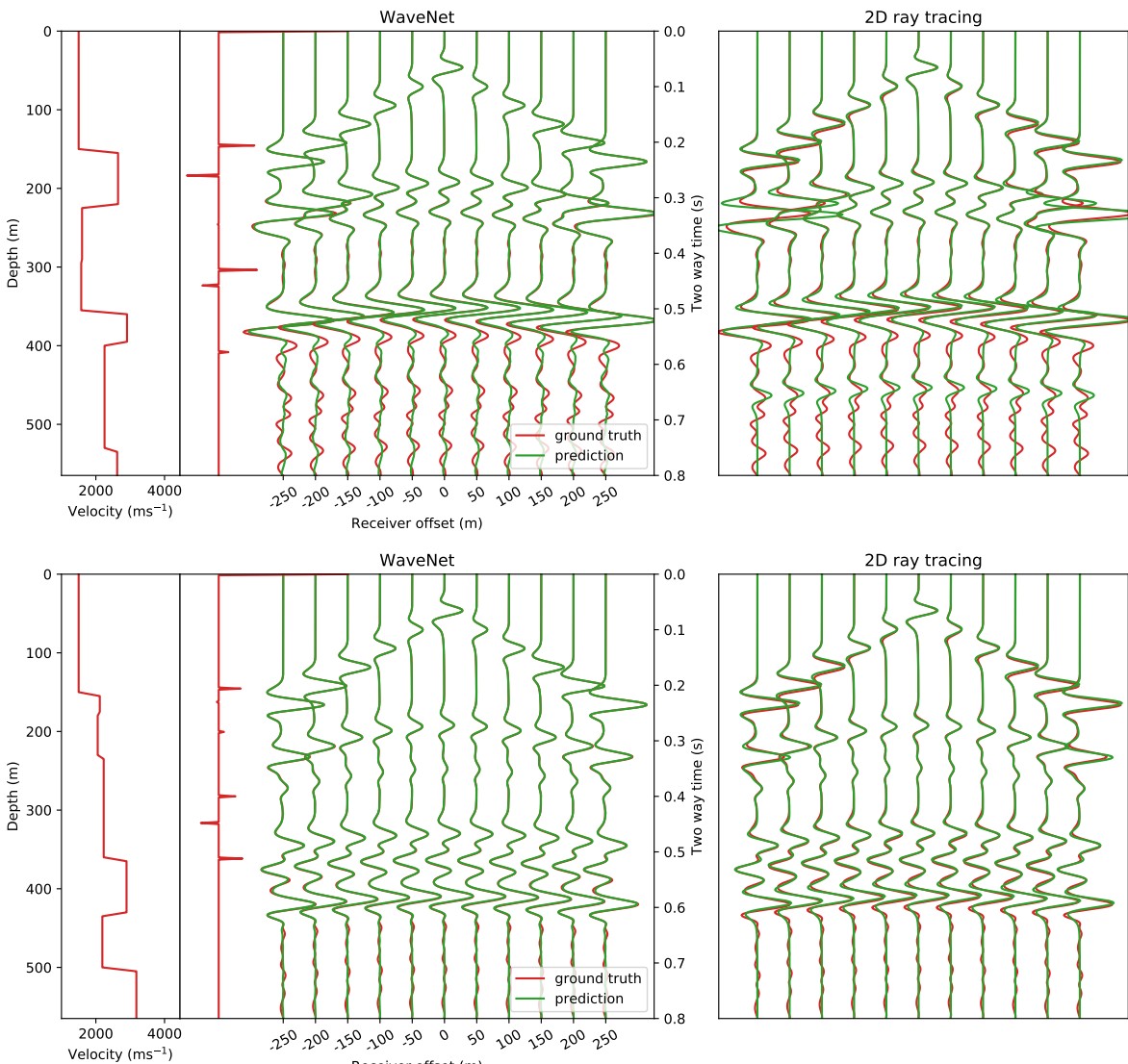

**Figure 5.** Comparison of WaveNet simulation to 2D ray tracing. We compare the WaveNet simulation to 2D ray tracing for 2 of the examples in Fig 4. Red shows the input velocity model, its corresponding reflectivity series and the ground truth pressure responses from FD simulation. Green shows the WaveNet simulation (left) and 2D ray tracing simulation (right). A $t^{2.5}$ gain is applied to the receiver responses for display.

honour time series prediction tasks which are causal and to better model input data which varies over multiple time scales. The WaveNet network recently achieved state of the art performance in text to speech synthesis.

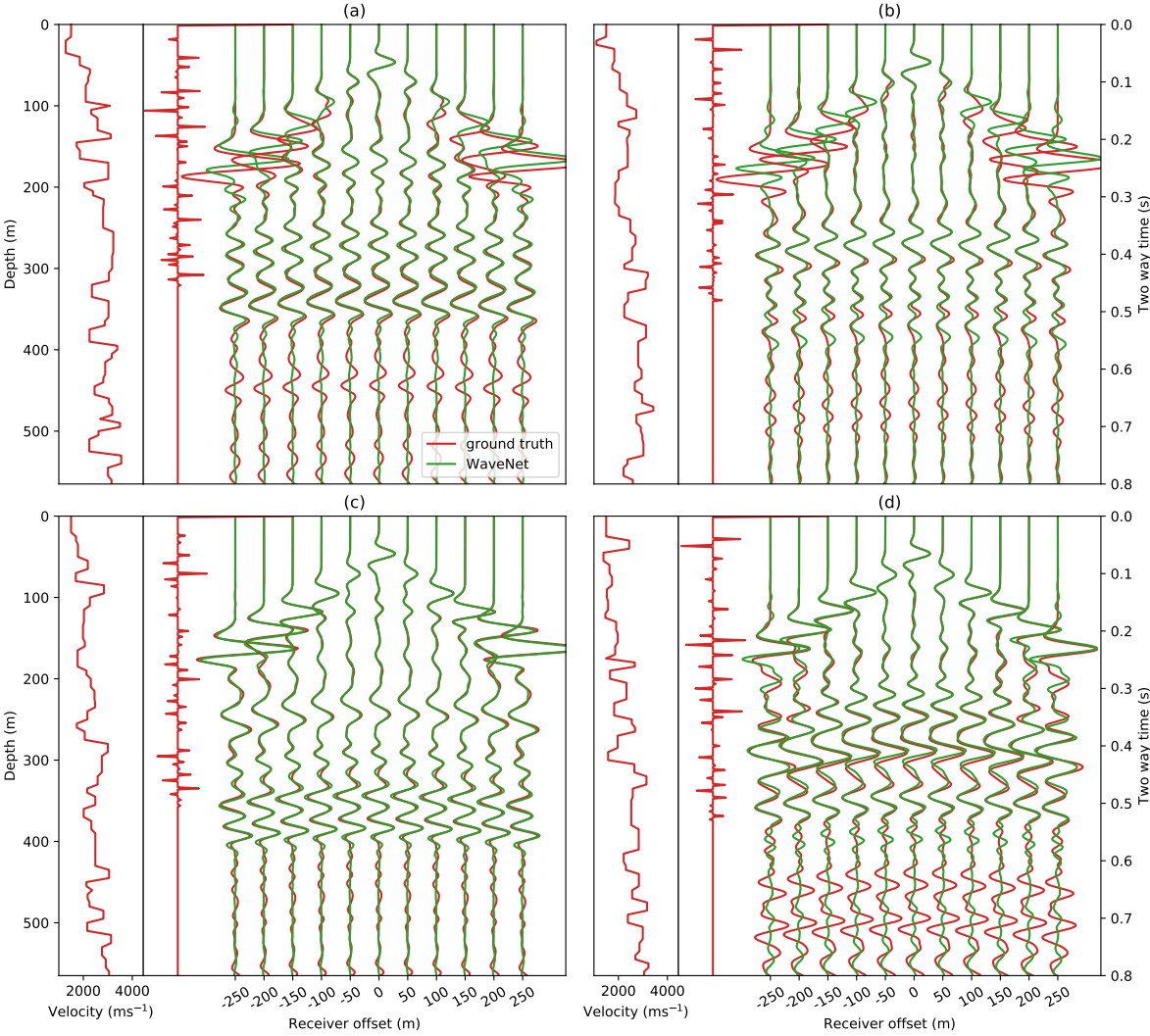

**Figure 6.** Generalisation ability of the WaveNet. The WaveNet simulations (green) for 4 velocity models with a much smaller average layer thicknesses than the training distribution are compared to ground truth FD simulation. Red shows the input velocity model, its corresponding reflectivity series and the ground truth pressure responses from FD simulation.

## 2.2 Simulation workflow

Our workflow consists of a preprocessing step, where we convert each input velocity model into its corresponding normal incidence reflectivity series sampled in time (Fig. 2, left), followed by a simulation step, where it is passed to a WaveNet network to simulate the pressure response recorded by each receiver (Fig. 2, middle).

The reflectivity series is typically used in exploration seismology (Russell, 1988) and contains values of the ratio of the amplitude of the reflected wave to the incident wave for each interface in a velocity model. For acoustic waves at normal incidence, these values are given by

$$R = \frac{\rho_2 v_2 - \rho_1 v_1}{\rho_2 v_2 + \rho_1 v_1} \; , \tag{2}$$

where $\rho_1, v_1$ and $\rho_2, v_2$ are the densities and P-wave velocities across the interface. The series is usually expressed in time and each reflectivity value occurs at the time at which the primary reflection of the source from the corresponding velocity interface arrives at a given receiver. The arrival times can be computed by carrying out a depth-to-time conversion of the reflectivity values using the input velocity model.

We chose to convert the velocity model to its reflectivity series and use the causal WaveNet architecture to constrain our 
workflow. For horizontally layered velocity models and receivers horizontally offset from the source, the receiver pressure recordings are causally correlated to the normal incidence reflectively series of the zero-offset receiver. Intuitively, a seismic reflection recorded after a short time has only travelled through a shallow part of the velocity model and the pressure responses are at most dependent on the past samples in this reflectivity series. By preprocessing the input velocity model into its corresponding reflectivity series and using the causal WaveNet architecture to simulate the receiver response we can constrain the 
network so that it honours this causal correlation.

We input the 1D profile of a 2D horizontally layered velocity model, with a depth of 640 m and a step size of 5 m. We use Eq. 2 and a standard 1D depth to time conversion to convert the velocity model into its normal incidence reflectivity series. The output reflectivity series has a length of 1 s and a sample rate of 2 ms. An example output reflectivity series is shown in Fig. 2 (left).

The reflectivity series is passed to the WaveNet network, which contains 9 causally-connected convolutional layers (Fig. 2, middle). Each convolutional layer has the same length as the input reflectivity series, 256 hidden channels, a receptive field width of 2 samples and a Rectified Linear Unit (ReLU) activation function (Nair and Hinton, 2010). Similar to the original WaveNet design, we use exponentially increasing dilations at each layer to ensure that the first sample in the input reflectivity series is in the receptive field of the last sample of the output simulation. We add a final causally-connected convolutional layer 
with 11 output channels, a filter width of 101 samples and an identity activation to generate the output simulation, where each output channel corresponds to a receiver prediction. This results in the network having 1,333,515 free parameters in total.

## 2.3 Training data generation

To train the network, we generate 50,000 synthetic ground truth example simulations using the SEISMIC_CPML code, which performs $2^{\text{nd}}$-order acoustic FD modelling (Komatitsch and Martin, 2007). Each example simulation uses a randomly sampled 
2D horizontally layered velocity model with a width and depth of 640 m and a sample rate of 5 m in both directions. (Fig. 1, bottom left). For all simulations we use a constant density model of $2200 \; \text{kgm}^{-2}$.

In each simulation the layer velocities and layer thickness are randomly sampled from log-normal distributions. We also add a small velocity gradient randomly sampled from a normal distribution to each model such that the velocity values tend

to increase with depth, to be more Earth-realistic. The distributions over layer velocities and layer thicknesses for the entire training set are shown in Fig. 3.

We use a 20 Hz Ricker source emitted close to the surface and record the pressure response at 11 receiver locations placed symmetrically around the source, horizontally offset every 50 m (Fig. 1, top left). We use a convolutional perfectly matched layer boundary condition such that waves which reach the edge of the model are absorbed with negligible reflection. We run each simulation for 1 s and use a 0.5 ms sample rate to maintain accurate FD fidelity. We downsample the resulting receiver pressure responses to 2 ms before using them for training.

We run 50,000 simulations and extract a training example from each simulation, where each training example consists of a 1D layered velocity profile and the recorded pressure response at each of the 11 receivers. We withhold 10,000 of these examples as a validation set to measure the generalisation performance of the network during training.

## 2.4 Training process

The network is trained using the Adam stochastic gradient descent algorithm (Kingma and Ba, 2014). This algorithm computes the gradient of a loss function with respect to the free parameters of the network over a randomly selected subset, or batch, of the training examples. This gradient is used to iteratively update the parameter values, with a step size controlled by a learning rate parameter. We propose a L2 loss function with time-varying gain function for this task, given by

$$L = \frac{1}{N} \|G(\hat{Y} - Y)\|_2^2 \,, \tag{3}$$

where $\hat{Y}$ is the simulated receiver pressure response from the network, $Y$ is the ground truth receiver pressure response from FD modelling and $N$ is the number of training examples in each batch. The gain function $G$ has the form $G = t^g$ where $t$ is the sample time and $g$ is a hyperparameter which determines the strength of the gain. We add this to empirically account for the attenuation of the wavefield caused by spherical spreading, by increasing the weight of samples at later times. In this Section we use a fixed value of $g = 2.5$. We use a learning rate of $1 \times 10^{-5}$, a batch size of 20 training examples and run training over 500,000 gradient descent steps.

## 2.5 Comparison to 2D ray tracing

We compare the WaveNet simulation to an efficient, quasi-analytical 2D ray-tracing algorithm which assumes horizontally layered media. We modify the 2D horizontally layered ray-tracing bisection algorithm from the CREWES seismic modelling library (Margrave and Lamoureux, 2018) to include Zoeppritz modelling of the reflection and transmission coefficients at each velocity interface (Aki and Richards, 1980) and 2D spherical spreading attenuation (Gutenberg, 1936; Newman, 1973) during ray tracing. The output of the algorithm is a primary reflectivity series for each receiver, which we convolve with the source signature used in FD modelling to obtain an estimate of the receiver responses.

## 2.6 Results

Whilst training the WaveNet the losses over the training and validation datasets converge to similar values, suggesting the network is generalising well to examples in the validation dataset. To assess the performance of the trained network, we generate a random test set of 1000 unseen examples. The simulations for 4 randomly selected examples from this test set are
compared to the ground truth FD modelling simulation in Fig. 4. We also compare the WaveNet simulation to 2D ray tracing in Fig. 5. For nearly all time samples the network is able to simulate the receiver pressure responses. The WaveNet is able to predict the Normal Moveout (NMO) of the primary layer reflections with receiver offset, the direct arrivals at the start of each receiver recording and the spherical spreading loss of the wavefield over time, though the network struggles to accurately simulate the multiple reverberations at the end of the receiver recordings.

We plot the histogram of the average absolute amplitude difference between the ground truth FD simulation and the simulation from the WaveNet and 2D ray tracing over the test set in Fig. A1 (bottom right) and observe that the WaveNet simulation has a lower average amplitude difference than 2D ray tracing. Small differences in phase and amplitude at larger offsets are the main source of discrepancy between the 2D ray tracing and FD simulation, which can be seen in Figure 5, and are likely due to errors both in the ray tracing approximation and in using discretisation in the FD simulation. The WaveNet predictions
are consistent and stable across the test set, and their closer amplitude match to the FD simulation is perhaps to be expected because the network is trained to directly match the FD simulation, rather than the 2D ray tracing.

We compare the sensitivity of the network's accuracy to two different convolutional network designs in Fig. A1. Their main differences to the WaveNet design is that both networks use standard rather than causal convolutional layers and the second network uses exponential dilations whilst the first does not. Both networks have 9 convolutional layers, each with 256 hidden
channels, filter sizes of 3, ReLU activations for all hidden layers and an identity activation function for the output layer, with 1,387,531 free parameters in total. We observe that the convolutional network without dilations does not converge during training, whilst the dilated convolutional network has a higher average absolute amplitude difference over the test set from the ground truth FD simulation than the WaveNet network (Fig. A1 (bottom right)).

The generalisation ability of the WaveNet outside of its training distribution is tested in Fig. 6. We generate four velocity
models with a much smaller average layer thickness than the training set and compare the WaveNet simulation to the ground truth FD simulation. We find that the WaveNet is able to make an accurate prediction of the seismic response, but it struggles to simulate the multiple reflections and sometimes the interference between the direct arrival and primary reflections.

We compare the average time taken to generate 100 simulations to FD simulation and 2D ray tracing in Table 1. We find that on a single CPU core the WaveNet is 19 times faster than FD simulation, and using a GPU and the TensorFlow library
(TensorFlow, 2015) it is 549 times faster. This speedup is likely to be higher than if the GPU was used for accelerating existing numerical methods (Rietmann et al., 2012). In this case, the specialised 2D ray tracing algorithm offers a similar speed up to the WaveNet network. The network takes approximately 12 hours to train on one Nvidia Tesla K80 GPU, although this training step is only required once and subsequent simulation steps are fast.

| Method | Average CPU time (s) | Average GPU time (s) | Training time (days) |
|---|---|---|---|
| 2D FD simulation | $73 \pm 1$ (1x) | - | - |
| 2D ray tracing | $2.2 \pm 0.1$ (33x) | - | - |
| WaveNet (forward) | $3.79 \pm 0.03$ (19x) | $0.133 \pm 0.001$ (549x) | 0.5 |
| Conditional autoencoder | $3.3 \pm 0.1$ (22x) | $0.180 \pm 0.003$ (406x) | 4 |
| WaveNet (inverse) | $1.27 \pm 0.02$ | $0.051 \pm 0.001$ | 0.5 |

**Table 1.** Speed comparison of simulation and inversion methods. The time shown is the average time taken to generate 100 simulations (or 100 velocity predictions for the inverse WaveNet) on either a single core of a 2.2 GHz Intel Core i7 processor or a Nvidia Tesla K80 GPU. For simulation methods the speed up factor compared to FD simulation is shown in brackets. The inverse WaveNet is faster than the forward WaveNet because it has less hidden channels in its architecture and therefore requires less computation.

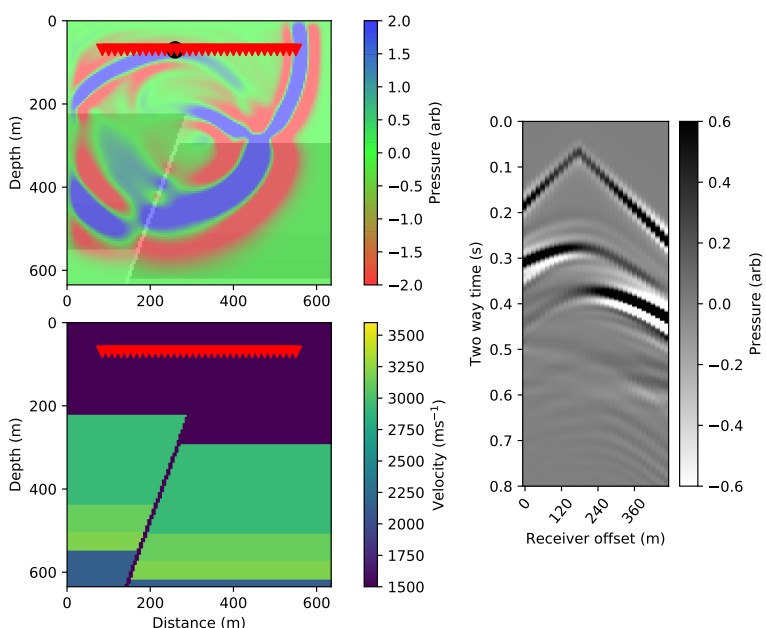

**Figure 7.** Ground truth FD simulation example, with a 2D faulted media. Left, top: The black circle shows the source location. 32 receivers are placed at the same depth as the source with a horizontal spacing of 15 m (red triangles). The full wavefield pressure is overlain for a single snapshot in time. Left, bottom: The Earth velocity model. Right: The resulting ground truth pressure response recorded by each receiver, using FD modelling. A $t^{2.5}$ gain is applied to the receiver responses for display.

## 3 Fast seismic simulation in 2D faulted acoustic media using a conditional autoencoder

The WaveNet architecture we implemented above is limited in that it is only able to simulate horizontally layered Earth models. In this section we present a second network which is significantly more general; it simulates seismic waves in 2D

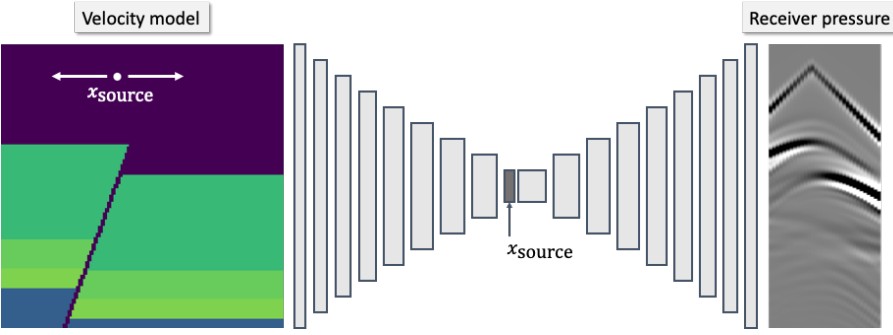

**Figure 8.** Our conditional autoencoder simulation workflow. Given a 2D velocity model and source location as input, a conditional autoencoder network outputs a simulation of the pressure responses at the receiver locations in Fig. 7. The network is composed of 24 convolutional layers and concatenates the input source location with its latent vector.

faulted acoustic media with arbitrary layers, fault properties and an arbitrary location of the seismic source on the surface of the media.

This is a much more challenging task to learn for multiple reasons. Firstly, the media varies along both dimensions and the resulting seismic wavefield has more complex kinematics than the wavefields in horizontally layered media. Secondly, we

allow the output of the network to be conditioned on the input source location which requires the network to learn the effect of the source location. Thirdly, we input the velocity model directly into the network without conversion to a reflectivity series beforehand; the network must learn to carry out its own depth to time conversion to simulate the receiver responses. We chose this approach over our WaveNet workflow because we note that for non-horizontally layered media the pressure responses are not causally correlated to the normal incidence reflectivity series in general and our previous causality assumption does not

hold.

Similar to Section 2, we simulate the seismic response recorded by a set of receivers horizontally offset from a point source emitted within the Earth model. An example simulation we wish to learn is shown in Fig. 7. We will now discuss the network architecture and training process in more detail below.

### 3.1 Conditional autoencoder architecture

Our simulation workflow is shown in Fig. 8. Instead of preprocessing the input velocity model to its associated reflectivity model, we input the velocity model directly into the network. The network is conditioned on the source position, which is allowed to vary along the surface of the Earth model. The output of the network is a simulation of the pressure responses recorded at 32 fixed receiver locations in the model shown in Fig. 7.

We use a conditional autoencoder network design, shown in Fig 8. The network is composed of 10 convolutional layers

which reduce the spatial dimensions of the input velocity model until it has a 1x1 shape with 1024 hidden channels. We term this tensor the latent vector. The input source surface position is concatenated onto the latent vector and 14 convolutional layers

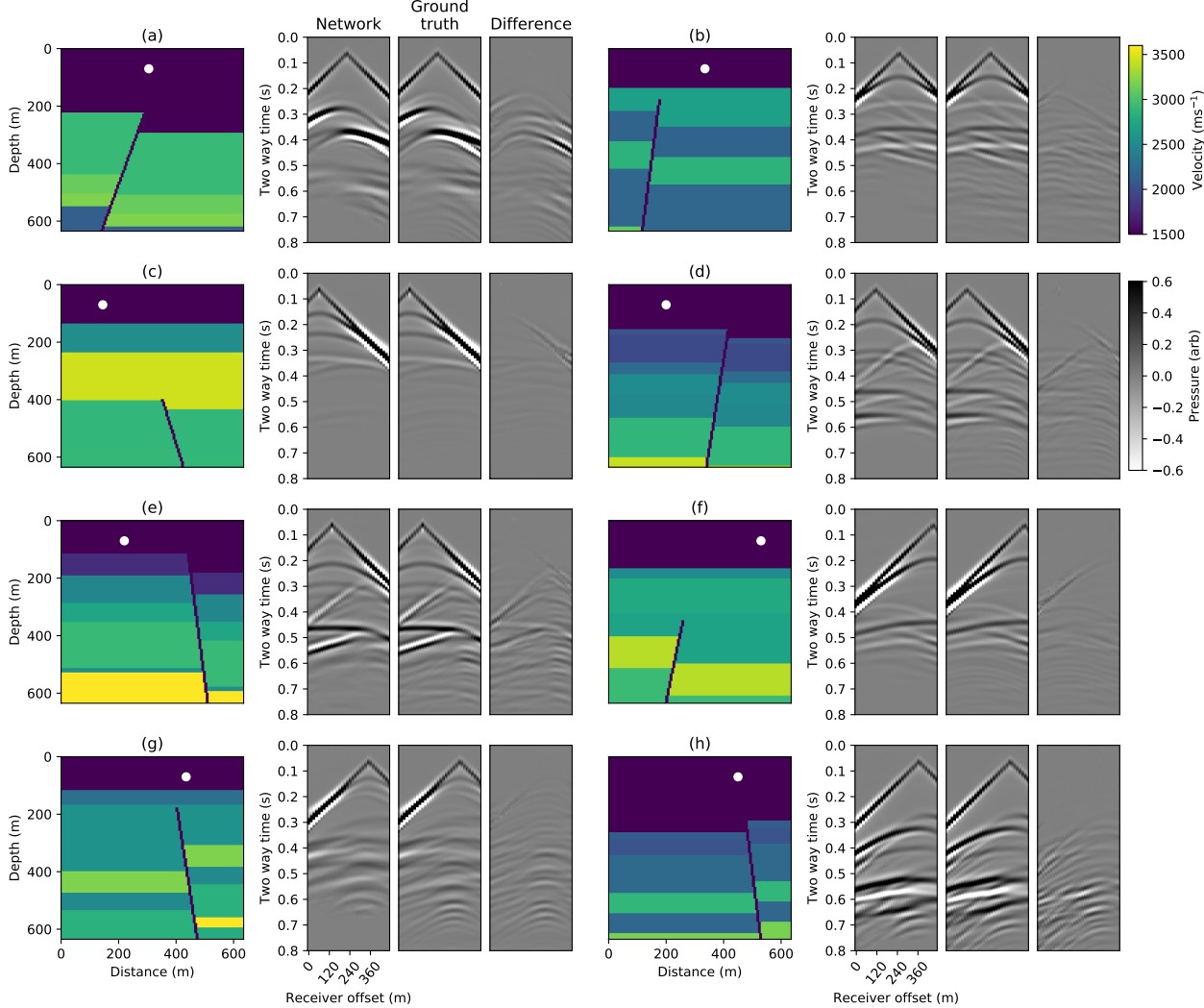

**Figure 9.** Conditional autoencoder simulations for 8 randomly selected examples in the test set. White circles show the input source location. The left simulation plots show the network predictions, the middle simulation plots show the ground truth FD simulations and the right simulation plots show the difference. A $t^{2.5}$ gain is applied for display.

are used to expand the size of the latent vector until its output shape is the same as the target receiver gather. We choose this encoder-decoder architecture to force the network to compress the velocity model into a set of salient features before expanding them to infer the receiver responses. All hidden layers use ReLU activation functions and the final output layer uses an identity activation function. The resulting network has 18,382,296 free parameters. The full parameterisation of the network is shown in Table A1.

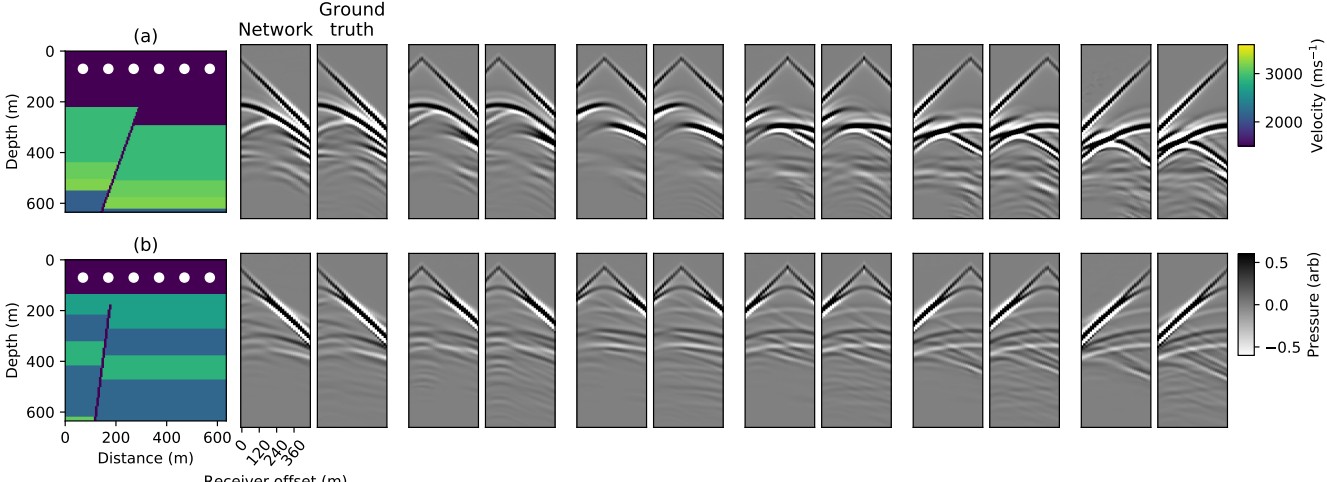

**Figure 10.** Conditional autoencoder simulation accuracy when varying the source location. The network simulation is shown for 6 different source locations whilst keeping the velocity model fixed. The source positions are regularly spaced across the surface of the velocity model (white circles). Example simulations for 2 different velocity models in the test set are shown, where each row corresponds to a different velocity model. The pairs of simulation plots in each row from left to right correspond to the network prediction (left in the pair) and the ground truth FD simulation (right in the pair), when varying the source location from left to right in the velocity model. A $t^{2.5}$ gain is applied for display.

## 3.2 Training process

We use the same training data generation process described by Section 2.3. When generating velocity models, we add a fault to the model. We randomly sample the length, normal or reverse direction, slip distance and orientation of the fault. Example velocity models drawn from this process are shown in Fig. 9. We generate 100,000 example velocity models and for each model chose three random source locations along the top of the model. This generates a total of 300,000 synthetic ground truth example simulations to use for training the network. We withhold 60,000 of these examples to use as a validation set during training.

   We train using the same training process and loss function described in Section 2.4, except that we employ a L1 norm instead of a L2 norm in the loss function (Eq. 3). We use a learning rate of $1\text{x}10^{-4}$, a batch size of 100 examples and run training over 3,000,000 gradient descent steps. We use batch normalisation (Ioffe and Szegedy, 2015) after each convolutional layer to help regularise the network during training.

## 3.3 Results

During training the losses over the training and validation datasets converge to similar values and we test the performance of the trained network using a test set of 1000 unseen examples. The output simulations for 8 randomly selected velocity models

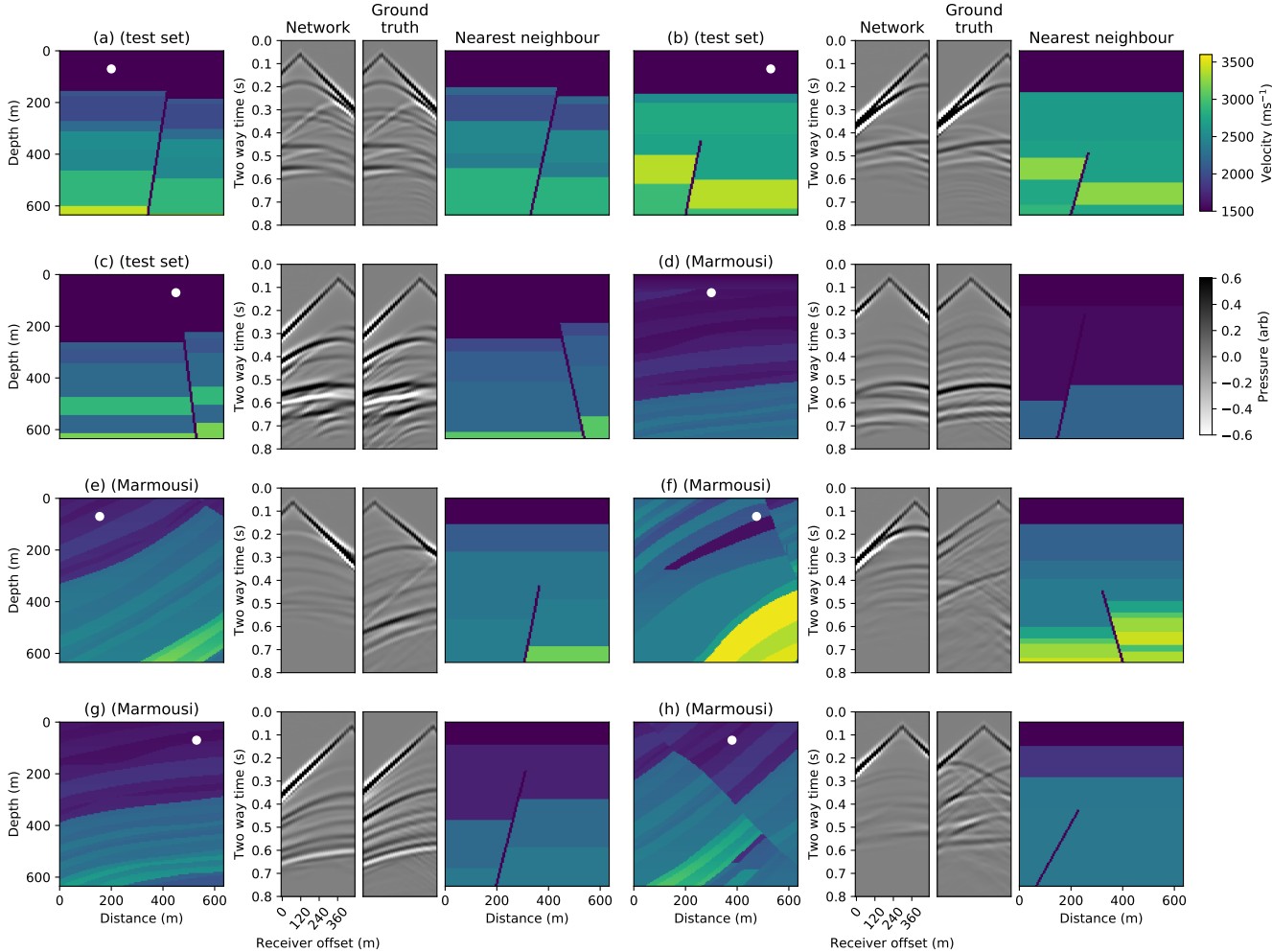

**Figure 11.** Generalisation ability of the conditional autoencoder. The conditional autoencoder simulations for 5 velocity models taken from different regions of the Marmousi P-wave velocity model are shown (examples (d)-(h)). For each example, left shows the input velocity model and source location, the middle simulations plots show the network prediction (left) and the ground truth FD simulation (right) and right shows the nearest neighbour in the training set to the input velocity model. Simulations from 3 of the test velocity models in Fig. 9 are also shown with their nearest neighbours (examples (a)-(c)). A $t^{2.5}$ gain is applied for display.

and source positions from this set are shown in Fig. 9. We observe that the network is able to simulate the kinematics of the primary reflections and in most cases is able to capture their relative amplitudes. We also plot the network simulation when varying the source location over 2 velocity models from the test set in Fig. 10 and find that the network is able to generalise well over different source locations.

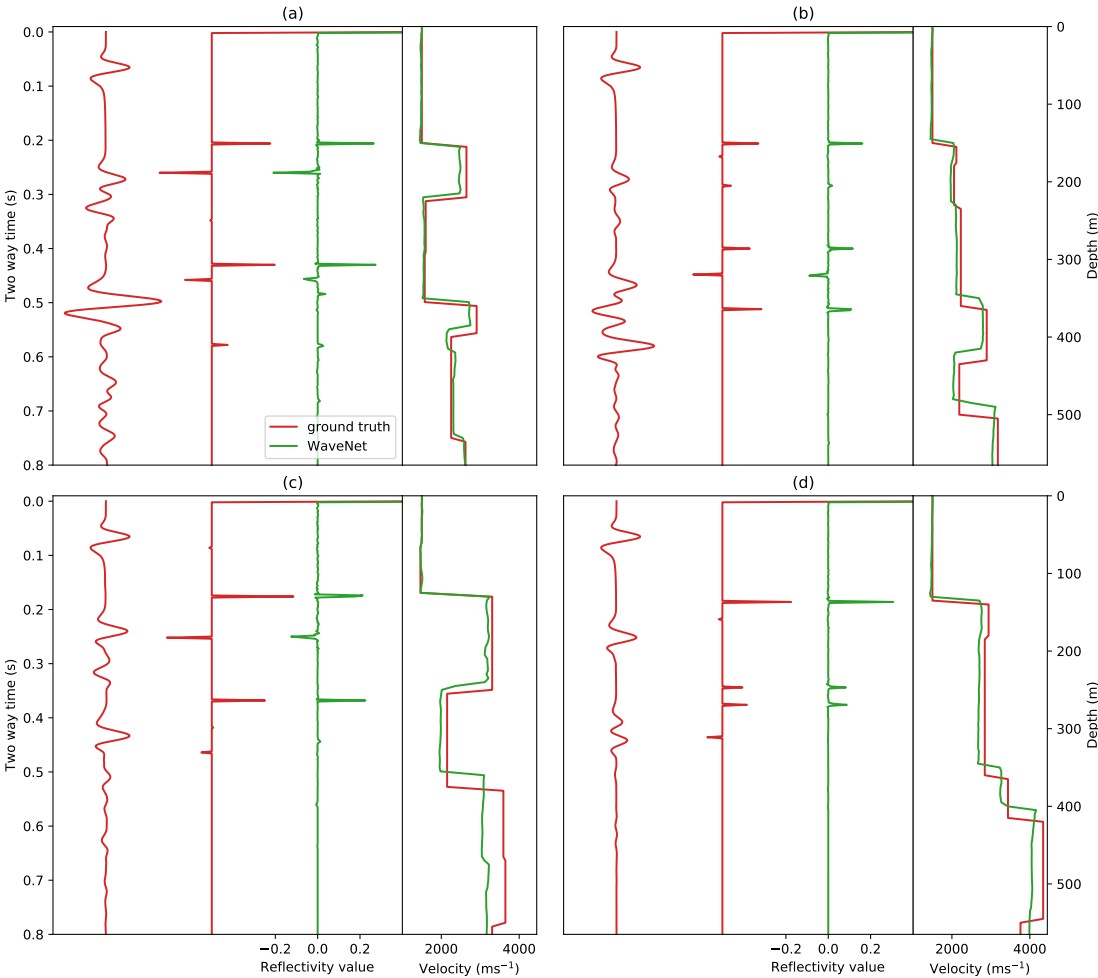

**Figure 12.** Top: Inverse WaveNet predictions for 4 examples in the test set. Red shows the input pressure response at the zero-offset receiver location, the ground truth reflectivity series and its corresponding velocity model. Green shows the inverse WaveNet reflectivity series prediction and the resulting velocity prediction.

We test the accuracy of the simulation when using different network designs and training hyperparameters, shown in Fig. A2. We compare example simulations from the test set when using our baseline conditional autoencoder network, when halving the number of hidden channels for all layers, when using an L2 loss function during training, when using gain exponents of $g = 0$ and $g = 5$ in the loss function and when removing 2 layers from the encoder and 8 layers from the decoder. We plot the histogram of the average absolute amplitude difference between the ground truth FD simulation and the network simulation over the test set for all of the cases above, and observe that in all cases the simulations are less accurate than our baseline approach. Without the gain in the loss function, the network only learns to simulate the direct arrival and the first few reflections in the receiver responses. With a gain exponent of $g = 5$, the network simulation is unstable and it fails to simulate

the first 0.2 seconds of the receiver responses. When using the network with less layers the simulations have edge artefacts, whilst the network with half the number of hidden channels is closest to the baseline accuracy. In testing we find that training a network with the same number of layers but without using a bottleneck design to reduce the velocity model to a 1x1x1024 latent vector does not converge.

We compare the accuracy of the conditional autoencoder to the WaveNet network in Fig. A3. We plot the simulation from both networks for an example model in the horizontally layered velocity model test set and the histogram of the average absolute amplitude difference between the ground truth FD simulation and the WaveNet and conditional autoencoder simulations over this set. Both networks are able to accurately simulate the receiver responses and the WaveNet simulation is slightly more accurate than the conditional autoencoder, though of course the latter is more general.

We test the generalisation ability of the conditional autoencoder outside of its training distribution by inputting randomly selected $640 \times 640$ m boxes from the publicly available 2D Marmousi P-wave velocity model (Martin et al., 2006) into the network. These velocity models contain much more complex faulting at multiple scales, higher dips and more layer variability than our training dataset. The resulting network simulations are shown in Fig. 11. We calculate the nearest neighbour to the input velocity model in the set of training velocity models, defined as the training model with the lowest L1 difference summed

over all velocity values from the input velocity model, and show this alongside each example.

      We find that the network is not able to accurately simulate the full seismic response from velocity models which have large dips and/or complex faulting (examples (e), (f) and (h)) that are absent in the training set. This observation is similar to most studies which analyse the generalisability of deep neural networks outside their training set (e.g. Zhang and Lin (2018) and Earp and Curtis (2020)). However, encouragingly, the network is able to mimic the response from velocity models with small

dips ((d) and (g)), even though the nearest training-set neighbour contains a fault whereas the Marmousi layers are continuous.

      We compare the average time taken to generate 100 simulations using the conditional autoencoder network to FD simulation in Table 1. We find that on a single CPU core the network is 22 times faster than FD simulation and when using a GPU and the PyTorch library (Pytorch, 2016) it is 406 times faster. This is comparable to the speed up obtained with the WaveNet. It is likely that 2D ray tracing will not offer the same speed up as observed in Section 2.6, because computing ray paths through

these models is likely to be more demanding. The network takes approximately 4 days to train on one Nvidia Titan V GPU. This is 8 times longer than training the WaveNet network, although we made little effort to optimise its training time. We find that when using only 50,000 training examples the validation loss increases and the network overfits to the training dataset.

## 4   Discussion

Both our deep neural networks accurately model the seismic response in horizontally layered and faulted 2D acoustic media.

The WaveNet is able to carry out simulation of horizontally layered velocity models, and the conditional autoencoder is able to generalise to faulted media with arbitrary layers, fault properties and an arbitrary location of the seismic source on the surface of the media. This is a significantly harder task than simulating horizontally layered media with the WaveNet network. Furthermore, both networks are one to two orders of magnitude faster than FD modelling.

Whilst these results are encouraging and suggest that deep learning is valuable for simulation, there are further challenges when extending our methods to more complex, elastic and 3D Earth models in practical simulation tasks. We believe that further research will help to understand whether deep learning can aid in these more general settings and discuss this in more detail below.

## 4.1 Extension to elastic simulation

An important ability for practical geophysical applications is to be able to simulate seismic waves in (visco-)elastic media, rather than acoustic media. The architectures of our networks are readily extendable in this regard; S-wave velocity and density models could be added as additional input channels to our networks and the number of output channels in the networks could be increased so that multi-component particle velocity vectors are output. The same training scheme could be used, with training data generated using elastic FD simulation instead of acoustic simulation and a loss function which compares vector fields instead of scalar fields. Thus, with some simple changes to our design, this challenge is at least conceptually simple to address, though further research is required to understand if it is feasible. The cost of traditional elastic simulation exceeds the cost of acoustic simulation by orders of magnitude and has prevented the seismic industry from fully embracing this crucial step. We postulate that the difference in simulation times between future elastic and acoustic simulation networks might be smaller compared to fully discretised methods such as FD, as a consequence of the networks not needing to compute the entire discretised wavefield. While this is speculative at this point, it is intriguing to investigate.

## 4.2 Extension to 3D simulation

Another important extension is to move from 2D to 3D simulation. In terms of network design, our autoencoder could be extended to 3D simulation by increasing the dimensionality of its input, hidden and output tensors. In this case we would expect a similar order of magnitude acceleration of simulation time to 2D, because the network would still directly estimate the seismic response without needing to iteratively model the seismic wavefield through time. However, multiple challenges arise in this setting. Firstly, increasing the dimensionality would increase the size of the network and therefore likely increase its training time. Finding an alternative representation, such as meshes or oct-trees (Ahmed et al., 2018) to reduce the dimensionality of the problem, or a way to exploit symmetry in the wave equation to reduce complexity, may be critical in this aspect. Secondly, a major challenge is likely to be the increased computational cost of generating training data with conventional methods, which for instance is significantly higher in 3D when using FD modelling. Whilst we only used the subset of the wavefield at each receiver location to train our networks, finding a way to use the entire wavefield from FD simulation to train the network may help reduce the number of training simulations required. We note that generating training data is an amortized cost because the network only needs to be trained once, and although large, in the case of seismic inversion where millions of production runs are required the training cost could become negligible. Another intriguing aspect is to investigate whether deep neural network simulation costs scale more favourably with increasing frequency $\omega$, compared to fully discrete methods which scale with $\omega^4$; in this study we only consider simulation at a fixed frequency range.

## 4.3 Generalisation to more complex Earth models

Perhaps the largest challenge in designing appropriate networks is to improve their generality so they can simulate more complex Earth models. We have shown that deep neural networks can move beyond simulating simple horizontally layered velocity models to more complex faulted models where, to the best of our knowledge, no analytical solutions exist, which we believe is a positive step. However, both our networks performed worse on velocity models outside of their training distributions. Furthermore, to be able to generalise to more complex velocity models the conditional autoencoder required more free parameters, more time to train and more training examples than the WaveNet network. Generalisation outside of the training distribution is a well known and common challenge of deep neural networks in general (Goodfellow et al., 2016).

A naive approach would be to increase the range of the training data to improve the generality of the network, however this would quickly become computationally intractable when trying to simulate all possible Earth models. We note that for many practical applications it may be acceptable to use a training distribution with a limited range; for example, in many of the seismic applications such tomography, FWI, and seismic hazard assessment, a huge number of forward simulations of comparatively few Earth models are carried out.

A promising research direction may be to better regularise the networks by adding more physics-based constraints into the workflow. We found that using causality in the WaveNet generated more accurate simulations than when using a standard convolutional network; this suggested that adding this constraint helped the network simulate the seismic response, although it is an open question how best to represent causality when simulating more arbitrary Earth models. We also found that a bottleneck design helped the conditional autoencoder to converge; our hypothesis is that this encouraged a depth-to-time conversion by slowly reducing the spatial dimensions of the velocity model before expanding them into time. More advanced network designs, for example using attention-like mechanisms (Vaswani et al., 2017) to help the network focus on relevant parts of the velocity model, rather than using convolutional layers with full fields of view, or using Long Short-Term Memory (LSTM) cells to help the network model multiple reverberations could be tested. Another interesting direction would be to use the wave equation (Eq. 1) to directly regularise the loss function, similar to the physics-based machine learning approach proposed by Raissi et al. (2019).

We find the nearest neighbour test is a useful way to understand if an input velocity model is close to the training distribution and therefore if the network's output simulation is likely to be accurate. Probabilistic approaches, such as Bayesian deep learning (Gal, 2016), could be investigated for their ability to provide more quantitative uncertainty estimates on the network's output simulation.

## 4.4 Inversion with WaveNet

As an additional test, we were also able to retrain the WaveNet network to carry out fast seismic inversion in the horizontally layered media, which offered a fast alternative to existing inversion algorithms. We retrained the WaveNet network with its inputs and output reversed; its input was then a set of 11 recorded receiver responses and its output was a prediction of the corresponding normal incidence reflectivity series. We used the same WaveNet architecture described in Section 2.2, except

that we inverted its structure to maintain the causal correlation between the receiver responses and reflectivity series, and we used 128 instead of 256 hidden channels for each hidden layer. We used exactly the same training data and training strategy described in Section 2.3 and 2.4, except that we used a loss function given by

$$L = \frac{1}{N}\|\hat{R} - R\|_2^2 , \tag{4}$$

5 where $R$ is the true reflectivity series and $\hat{R}$ is the predicted reflectivity series. To recover a prediction of the velocity model we carried out a standard 1D time-to-depth conversion of the output reflectivity values followed by integration.

Predictions of the reflectivity series and velocity models for 4 randomly selected examples from a test set of unseen examples are shown in Fig. 12. The inverse WaveNet network was able to predict the underlying velocity model for each example, although in some cases small velocity errors propagated with depth, which was likely a result of the integration of the reflectivity 10 series. The network was able to produce velocity predictions in the same order of magnitude time as the forward network (shown in Table 1), which is likely to be a fraction of the time needed for existing seismic inversion algorithms which rely on forward simulation.

We note that seismic inversion is typically an ill-defined problem and it is likely that the predictions of this network are biased towards the velocity models it was trained on. We expect the accuracy of the network to reduce when tested on inputs 15 outside of its training distribution and with real, noisy seismic data. Further research could try to quantify this uncertainty, for example by using Bayesian deep learning. We have not yet compared our inverse WaveNet network to existing seismic inversion techniques, such as posterior sampling or FWI.

An alternative method for inversion is to use our forward networks in existing seismic inversion algorithms based on optimisation, such as FWI. Both the WaveNet and conditional autoencoder networks are fully differentiable and could therefore 20 be used to generate fast approximate gradient estimates in these methods. However, similar limitations on their generality are likely to exist and one would need to be careful to keep the inversion routine within the training distribution of the network. Furthermore, whilst fast, these approaches would still suffer from curse of dimensionality when moving to higher dimensions, and require exponentially more samples to fully explore the parameter space.

### 4.5 Summary

25 Given the potentially large training costs and the challenge of generality, it may be that current deep learning techniques are most advantageous to practical simulation tasks where many similar simulations are required, such as inversion or statistical seismic hazard analysis, and least useful for problems with a very small number of simulations per model family. In seismology, however, we suspect that most current and future challenges fall into the former category, which renders these initial results promising. Deep learning approaches have different computational costs and benefits, and accuracies that are less clearly 30 understood compared to traditional approaches and these should be considered for each application. Further research is required to understand how best to design the training set for a particular simulation application, as well as how to help deep neural networks generalise to unseen velocity models outside of the training distribution. Finally we note that we only tested two

types of deep neural networks (WaveNet and conditional autoencoders) and many other types exist which could prove more effective.

## 5  Conclusions

We have investigated the potential of deep learning for aiding seismic simulation tasks in geophysics. We presented two deep neural networks which are able to carry out fast and largely accurate simulation of seismic waves. Both networks are 20 - 500 times faster than FD modelling and simulate seismic waves in horizontally layered and faulted 2D acoustic media. The first network uses a WaveNet architecture and simulates seismic waves in horizontally layered media. We showed that this network can also be used to carry out fast seismic inversion of the same media. The second network is significantly more general than the first; it simulates seismic waves in faulted media with arbitrary layers, fault properties and an arbitrary location of the seismic source on the surface of the media. Our main contribution is to show that deep neural networks can move beyond simulating simple horizontally layered velocity models to more complex faulted models where, to the best of our knowledge, no analytical solutions exist, which we believe is a positive step towards understanding their practical potential. We discussed the challenges of extending our approaches to practical geophysical applications and future research directions which could address them, noting where it may be favourable for using these network architectures.

## Appendix A:  Supplementary figures and tables

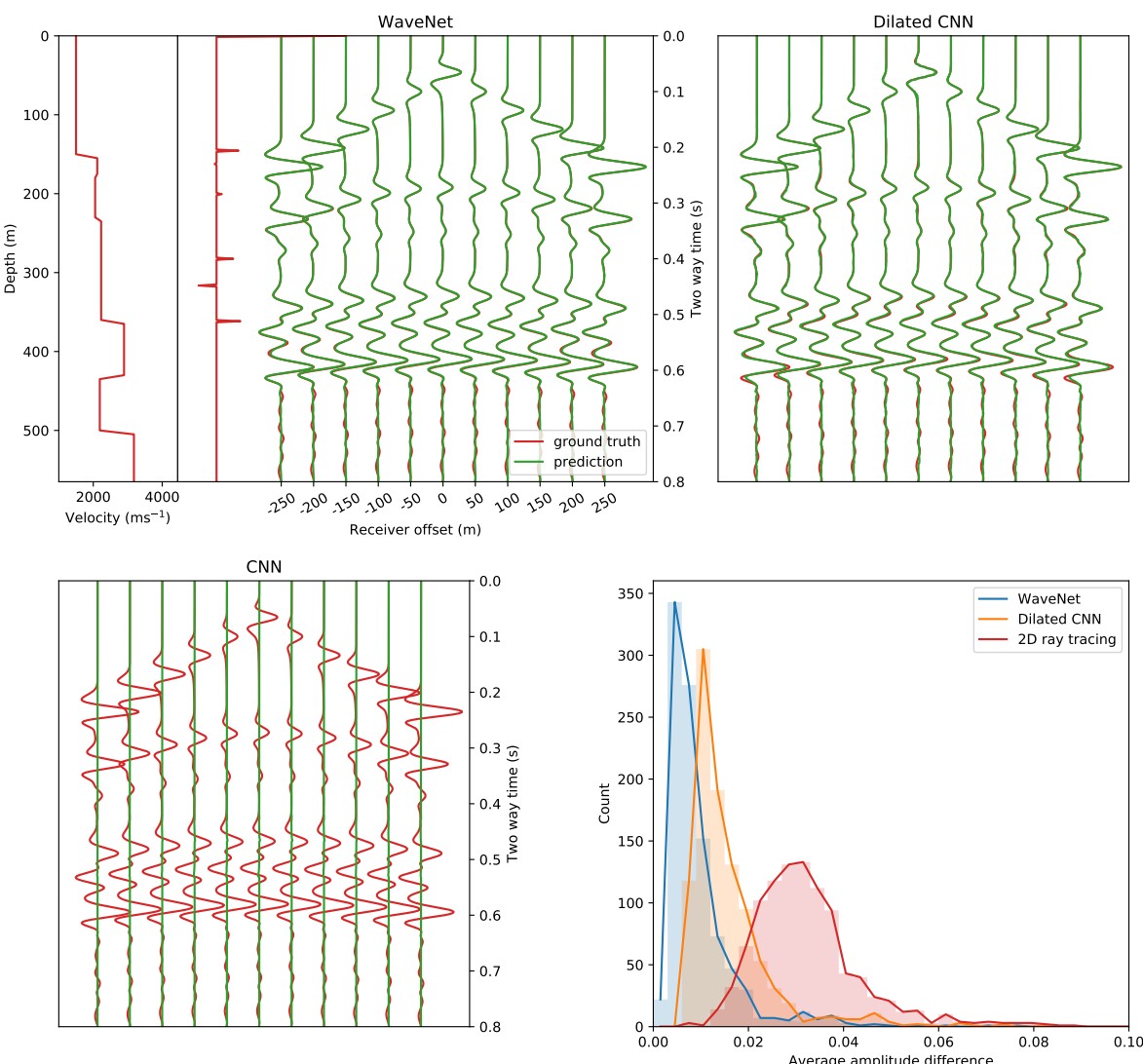

**Figure A1.** Comparison of different network architectures on simulation accuracy. Top left shows the WaveNet simulated pressure response for a randomly selected example in the test set (green) compared to ground truth FD simulation (red). Top right and bottom left show the simulated response when using two convolutional network designs with and without exponential dilations. Bottom right shows the histogram of the average absolute amplitude difference between the ground truth FD simulation and the simulations from the WaveNet, the dilated convolutional network and 2D ray tracing over the test set of 1000 examples. A $t^{2.5}$ gain is applied to the receiver responses for display.

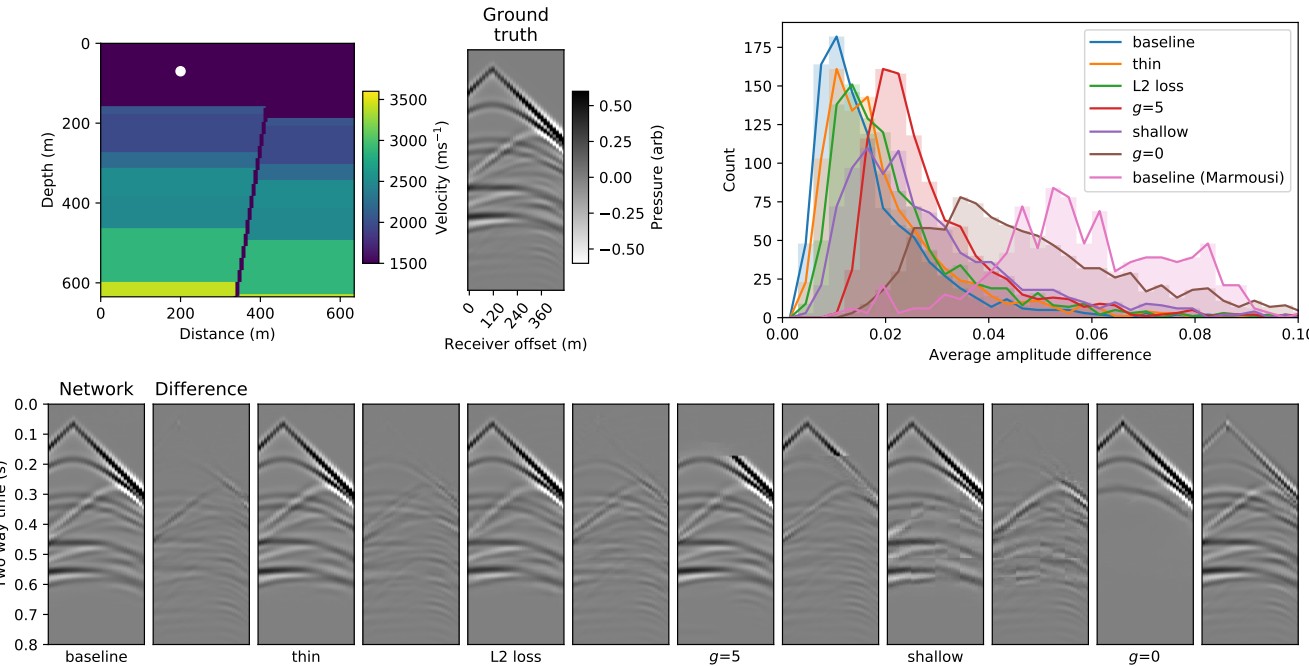

**Figure A2.** Comparison of different conditional autoencoder network designs and training hyperparameters on simulation accuracy. Top left shows a randomly selected velocity model and source location from the test set and its corresponding ground truth FD simulation. Bottom compares simulations and their difference to the ground truth when using our proposed conditional autoencoder (baseline), when halving the number of hidden channels for all layers (thin), when using an L2 loss function during training (L2 loss), when using gain exponents of $g = 0$ and $g = 5$ in the loss function and when removing 2 layers from the encoder and 8 layers from the decoder (shallow). Top right shows the histogram of the average absolute amplitude difference between the ground truth FD simulation and the simulation from the different cases over the test set. The histogram of the baseline network over the Marmousi test dataset is also shown. A $t^{2.5}$ gain is applied for display.

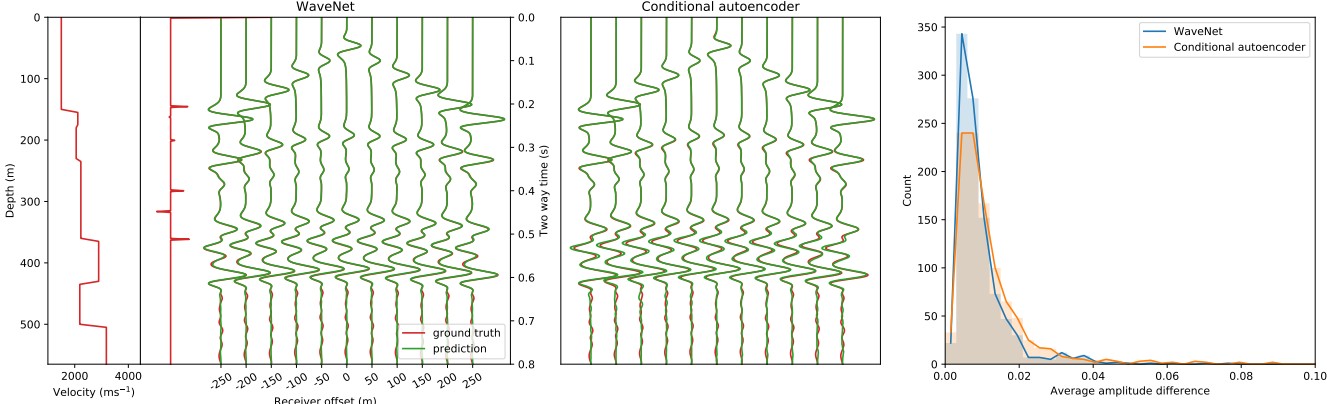

**Figure A3.** Comparison of the WaveNet and conditional autoencoder simulation accuracy. The left plot shows a velocity model, reflectivity series and ground truth FD simulation for a randomly selected example in the horizontally layered velocity model test set in red. Green shows the WaveNet simulation. The middle plot shows the conditional autoencoder simulation for the same velocity model. The right plot shows the histogram of the average absolute amplitude difference between the ground truth FD simulation and the WaveNet and conditional autoencoder simulations over this test set. A $t^{2.5}$ gain is applied for display.

| Layer | Type | in, out channels | kernel size | stride | padding | | | | | | |
|-------|------|-----------------|-------------|--------|---------|----|---------|-------------|-------|-------|-------|
| 1 | Conv2d | (1,8) | (3,3) | (1,1) | (1,1) | 14 | Conv2d | (512,512) | (3,3) | (1,1) | (1,1) |
| 2 | Conv2d | (8,16) | (2,2) | (2,2) | 0 | 15 | Conv2d | (512,512) | (3,3) | (1,1) | (1,1) |
| 3 | Conv2d | (16,16) | (3,3) | (1,1) | (1,1) | 16 | ConvT2d | (512,256) | (2,4) | (2,4) | 0 |
| 4 | Conv2d | (16,32) | (2,2) | (2,2) | 0 | 17 | Conv2d | (256,256) | (3,3) | (1,1) | (1,1) |
| 5 | Conv2d | (32,32) | (3,3) | (1,1) | (1,1) | 18 | Conv2d | (256,256) | (3,3) | (1,1) | (1,1) |
| 6 | Conv2d | (32,64) | (2,2) | (2,2) | 0 | 19 | ConvT2d | (256,64) | (2,4) | (2,4) | 0 |
| 7 | Conv2d | (64,128) | (2,2) | (2,2) | 0 | 20 | Conv2d | (64,64) | (3,3) | (1,1) | (1,1) |
| 8 | Conv2d | (128,256) | (2,2) | (2,2) | 0 | 21 | Conv2d | (64,64) | (3,3) | (1,1) | (1,1) |
| 9 | Conv2d | (256,512) | (2,2) | (2,2) | 0 | 22 | ConvT2d | (64,8) | (2,4) | (2,4) | 0 |
| 10 | Conv2d | (512,1024) | (2,2) | (2,2) | 0 | 23 | Conv2d | (8,8) | (3,3) | (1,1) | (1,1) |
| 11 | Concat | (1024,1025) | | | | 24 | Conv2d | (8,8) | (3,3) | (1,1) | (1,1) |
| 12 | ConvT2d | (1025,1025) | (2,2) | (2,2) | 0 | 25 | Conv2d | (8,1) | (1,1) | (1,1) | 0 |
| 13 | ConvT2d | (1025,512) | (2,4) | (2,4) | 0 | | | | | | |

**Table A1.** Conditional autoencoder layer parameters. Each entry shows the parameterisation of each convolutional layer. The padding column shows the padding on each side of the input tensor for each spatial dimension.

*Code and data availability.* All our training data used was generated synthetically, using the SEISMIC_CPML FD modelling library. Our WaveNet code is already publicly available on Github here: https://github.com/benmoseley/seismic-simulation-wavenet. We are happy to release the code to reproduce all our results on Github on publication of this paper.

*Author contributions.* TNM and AM were involved in the conceptualisation, supervision and review of the work. BM was involved in the conceptualisation, data creation, methodology, investigation, software, data analysis, validation and writing.

*Competing interests.* TNM is a Topical Editor for the Solid Earth Editorial Board.

*Acknowledgements.* The authors would like to thank the Computational Infrastructure for Geodynamics (www.geodynamics.org) for releasing the open-source SEISMIC_CPML FD modelling libraries.

We would also like to thank Tom Le Paine for his fast WaveNet implementation on GitHub from which our code was based on (github.com/tomlepaine/fast-wavenet).

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
