# Peer review of "Deep learning for fast simulation of seismic waves in complex media"

_Solid Earth, 2019_

## Referee Comment (RC1) · Andrew Valentine (Referee) · 22 Dec 2019

This paper by Moseley et al. describes how modern deep learning approaches can be used to construct a fast approximation to a seismological forward modelling code. This is an interesting and timely contribution, and the manuscript itself is clear and well-written. I have a few comments, set out below, but I have no hesitation in recommending the manuscript for acceptance once the authors have had an opportunity to respond to these.

- In any ML-based approach, the training data is central to the applicability of the method. The author's trained network appears effective for simulating waveforms in models that are generated using the same criteria as were used to make the

training set. However, I suspect performance will be significantly worse for models that have significantly different character. This is something that deserves more discussion than it receives, perhaps with some examples. A particular concern in practical settings may be how an end-user can assess whether their input model is 'sufficiently close' to the training set.

- The main 'selling point' of the author's approach is that it enables seismograms to be generated significantly faster than would be possible using 'traditional' forward models. However, this comes with a number of caveats that I think need to be discussed more carefully.

    1. (As above) the author's approach is (I suspect) only effective for models that are sufficiently similar to those in the training set. The numerical forward code does not suffer from this restriction, and can handle complexities that aren't present in the authors' setup (e.g. anisotropy, variable density). How much of a speedup could be achieved by using a numerical code that had been designed with prior knowledge of the characteristics of the authors' training set? Put another way: the speedup could be made to seem even more impressive by using a code designed for a vastly more complex setting (e.g. SPECFEM3D) to build the training set. How fair is the comparison that is being presented?

    2. The headline speed comparisons ignore the costs of building a training set and then training the neural networks, which are significant. How many simulations does a user need to envisage performing before the author's approach becomes cost-effective overall? I think this is going to be a rather large number. Again, some discussion of the pros and cons of the author's approach would seem desirable.

- What do the authors foresee as the primary application(s) of their approach? The discussion seems to mainly envisage inversion-related use cases. Some

comments on this:

1. The 'fast seismic inversion' approach discussed in Section 3 is essentially a variant of the 'prior sampling' approach discussed in detail by Käufl et al (2016). The key strength and weakness of this as an inversion strategy is that all samples (i.e. the training set) are generated without reference to any observed data. This enables very rapid inference once data becomes available, but it means that most training samples lie far from the observed data and are largely wasted from the perspective of any one inference. The end result is that inferences are considerably less well-constrained than would be possible with posterior sampling (see Fig. 9 of Käufl et al 2016). The bottom line is that prior sampling only seems a worthwhile strategy for (a) problems where time is of the essence, e.g. earthquake early warning, or (b) problems where the 'same' inference task needs to be solved many thousands of times with different data vectors.

2. Using the learned model in Monte Carlo simulations seems superficially attractive, but comes with significant caveats. Fundamentally the inference remains entirely based on the information contained within the training dataset, and so all the limitations of prior sampling remain. The random walk would need to be constrained to only generate models compatible with the training data, if results are to be meaningful. Perhaps it would be possible to progressively retrain the learned simulation as the McMC proceeds, to ensure accuracy in relevant parts of the model space: this starts to move towards the Bayesian optimisation approaches discussed in (e.g.) Wang et al (2013). To play devil's advocate: if a problem is too complex to tackle using an McMC approach using physical simulations, can we really be confident that a learned model is sufficiently accurate to yield meaningful results? How big a training set is required to capture the full complexity of the physical problem?

Some discussion and commentary on these issues, and other potential applications, would be appreciated.

- In general, we can make numerical simulations faster by introducing physical approximations. In such cases we typically have some intuition for how those approximations will impact upon results. Learned models offer a speedup without explicit physical approximations, but come with uncertainties that are difficult to quantify rigorously, and which may vary considerably depending on the particular set of inputs chosen. Would the authors like to comment on the pros and cons of the two different strategies for reducing computational costs?

- Referencing, especially in the introduction, seems rather haphazard. If citations are to be given for broad, well-established topics such as the utility of seismic simulations in reservoir characterisation, I would expect these to be to major review papers or to 'classics': these are going to be most useful for a reader who is unfamiliar with the field. Without intending any criticism of the cited works, this does not really seem to be the case at present. Moreover, the authors' survey of the history of machine learning in geophysics is very short-sighted, ignoring anything more than a couple of years old. There are neural network papers in the geophysical literature from the late 1970s onwards, and it would be nice to see some acknowledgement of this body of work. Valentine & Trampert (2012) is probably the first instance of 'deep learning' in seismology, though the term had not been invented at that point (and we did not have the benefit of modern computational frameworks).

**References**:

Käufl et al, 2016. Solving probabilistic inverse problems rapidly with prior samples. https://academic.oup.com/gji/article/205/3/1710/656483

Valentine & Trampert, 2012. Data space reduction, quality assessment

and searching of seismograms: autoencoder networks for waveform data. https://academic.oup.com/gji/article/189/2/1183/622660

Wang et al, 2013. Bayesian Optimization in a Billion Dimensions via Random Embeddings. https://arxiv.org/abs/1301.1942
* * *

---

## Referee Comment (RC2) · Andrew Curtis (Referee) · 24 Dec 2019

*Review of Solid Earth - Discussions manuscript:*

**Deep learning for fast simulation of seismic waves in complex media**

by Ben Mosely, Tarje Nissen-Meyer and Andrew Markham

*Review by Andrew Curtis, University of Edinburgh, UK.*

**Summary:**

This manuscript presents interesting work which attempts to create useful methods for modelling and inverting acoustic waveforms for 1D (depth-varying) media, with or without the addition of an offset fault. It tests two architectures of neural network and shows that they are capable of modelling the waveforms to within a small error tolerance relative to the data magnitude, and contributes towards showing that similar networks might be used to invert recorded waveform data.

To my mind this manuscript presents really interesting results in the field of acoustic modelling of layered media, and a taster of potential future results for waveform inversion. It certainly makes a contribution to scientific research. However, in its current form it does not explain how it contributes to **Solid Earth** research. The manuscript also lacks an overview/introduction/review/discussion of much past research, and as such it is not clear whether the authors are aware of whether or how their work improves on past work in the area of Geophysics/solid earth research. The latter comment is particularly apparent with respect to waveform inversion: published work from the 1990's appears to achieve almost as much as this paper (this may not be true, but the authors need to explain why).

I also agree with most (but not all) of the comments from Reviewer 1 (Andrew Valentine), in particular his comment that a fair comparison for the layered-medium example would be between neural networks and other modelling methods that intrinsically assume a 1D Earth structure rather than full finite difference methods. I will therefore not repeat his comments here. However, I outline below a couple of places where I would perhaps add to his comments.

Best wishes to the Editor, Reviewer 1 and to the Authors,

Andrew Curtis.

**Comments:**

1. The main worry I have about this manuscript is a combination of (a) the method is acoustic and uses relatively simple acoustic models, (b) the authors themselves state that it will be difficult to extend the proposed methods to more complex models and particularly to solid (elastic) models, and (c) that these two points seem to imply that this method is not, as the authors propose, an advancement on solid Earth science but rather is a paper about toy problems in acoustics that can not be extended to elastic media. I am not saying that this is definitely what the authors believe, nor that it is true, but that is how their message came across to me as a reader. If true (and I would not actually be surprised if it is), this suggests that the paper might be more appropriate for an acoustics journal like JASA, rather than a solid Earth journal. Otherwise, the authors must better justify how this work advances solid Earth science.

It may be that the authors think that these methods could be extended to elastic media and to significantly more complex (heterogeneous) models, and they simply have not explained how. Or it may be that they think that this research closes off an avenue of research for solid Earth science which is useful to stop others from following – in effect they might decide to argue that they prove that this approach or network architecture will not be fruitful for the Earth sciences. Either is a positive step for science. One way or the other, the authors need to explain more clearly which (or which other message) we should take from their work, and why.

However, since this is a Discussion paper, in fact I think the best would be to rethink the paper slightly: I would begin by thinking through, and then presenting, a roadmap that *might* solve real modelling or inversion problems in the solid Earth sciences using models of more realistic complexity. Then show how this paper fits into that – either taking a first step in a possible direction to achieve it, or testing an avenue that while successful, turns out not to be practical for the real Earth. Either way, in the Discussion section they can explain how this work has advanced our state of knowledge about the overall strategy, and how we should move forward in future.

2.  The introduction is interesting and reviews some of the appropriate material, but is very sparsely justified, and as also stated by Reviewer 1, it does not include many key references. In my view, every sentence of a scientific work must either be a logical deduction from previous text, must have been deduced/proven in another paper, or may be an argument based on the material in another paper; in that latter two cases that paper needs to be discussed and cited. In this paper, the Introduction cites very few references and therefore contains unjustified (in the sense of, 'not justified') statements. Examples include:

    One cannot write a paper on using neural networks to perform full waveform inversion (FWI) without citing Roeth and Tarantola (1994 – J. Geophys. Research). How does the FWI part of this paper improve on their work? That is not at all clear. There are many other papers using neural networks for imaging in Geophysics using waveforms or other types of data; you need to read and cite them, and describe how this work advances the field relative to those works. Currently the latter is not clear. The authors make the case for using neural networks for real-time applications – again first steps in this direction have already been taken (see Cao et al., 2019 – Geophysics, for example) and should be discussed.

3.  The authors promote the fact that they use 'deep learning', and their application certainly fits into that category. However, they must at least discuss why this is a positive feature of the method, and cite previous Geophysical applications of deep learning to support that discussion. Deep learning is usually defined to be the use of 4 or more layers within a neural network. While I agree with Reviewer 1 that his previous work (Valentine and Trampert) was an example of deep learning, the first that I know of in Geophysics was in fact Devilee et al., (1999 – J. Geophys. Res) – which came from the same university.

    In my view there is therefore nothing new about the concept of deep learning: we were using it in Geophysics in the '90's. What *has* changed is the extent to which depth can be used to impose useful structure on networks (as the authors themselves have done in this manuscript – their Figure 9, and also in the paper cited by Reviewer 1); also the number of parameters that can now be used (the width of each layer) has increased hugely. In fact the number of parameters in the authors' application is relatively modest compared to some in machine learning literature, but is

certainly comparable to other recent studies in Geophysics; the structure that the authors impose is both sensible and clearly useful in order to help to obtain stable results. These things should be discussed.

4. The paper appears to have committed the equivalent of an 'inverse crime'. If I understand correctly, the authors have trained networks in the forward and inverse directions using models with a certain parametrisation, and have tested the networks on models of exactly the same parametrization (Reviewer 1 touched on this too). While this would be reasonable if the models were themselves reasonably realistic, in a practical field like Geophysics I think it is necessary to test the networks on examples that lie outside of the range of the training set – not only using different models from those in the training set, but models that are not within the span of the algorithm used to generate the training set (as in the real Earth).

   For example, (a) Earp and Curtis (2019, arXiv – https://arxiv.org/abs/1907.00541 ) and (b) Earp et al. (2019, arXiv – https://arxiv.org/abs/1908.09588 ) perform (a) probabilistic travel time tomographic imaging, and (b) probabilistic surface wave inversion for averaged shear wave structures with depth, using deep neural networks. The test examples using in both cases are created using a finer parametrization than was used in the network training set – thus the actually structure of the (synthetically) 'true' Earth is not attainable by the networks; nevertheless, they can be used as a useful check of whether in such cases (as in the real Earth) the networks behave sensibly – giving results that are spatial averages in some sense of the 'true' structure. To be clear I do not think that the above references are perfect in this regard and could certainly be improved (e.g., could use even more complex models for tests); nevertheless the authors could usefully think about such tests for their work as it would strengthen the conclusions.

Although the above may be read as being rather critical, I would like to be clear that I do like this paper, and I think that it could create a usefully contribution to the field and be accepted for publication in Sold Earth. I just think that as it stands now, it is framed as a useful contribution to practical Solid Earth Science, yet it does not explain whether or how that is achieved. It may be that only a reframing of the paper is necessary, but I wonder whether a little more actual research is needed too in order to fulfill the papers own promise.

---

## Author Comment (AC1) · 25 Jan 2020

**Author response to referee comments**

**Deep learning for fast simulation of seismic waves in complex media**

**Introduction:**
This document contains our responses and proposed manuscript changes resulting from the two reviews of our "Deep learning for fast simulation of seismic waves in complex media" discussion paper.

We would like to thank the reviewers (Andrew Curtis and Andrew Valentine) for their in-depth and valuable comments on the paper and hope our responses below address their comments.

Where we have deemed appropriate, we have grouped similar comments from both reviewers together and provided a single response.

Ben Moseley, Tarje Nissen-Meyer & Andrew Markham

Reviewer comments:

*[Reviewer 2] 1. The main worry I have about this manuscript is a combination of (a) the method is acoustic and uses relatively simple acoustic models, (b) the authors themselves state that it will be difficult to extend the proposed methods to more complex models and particularly to solid (elastic) models, and (c) that these two points seem to imply that this method is not, as the authors propose, an advancement on solid Earth science but rather is a paper about toy problems in acoustics that cannot be extended to elastic media. I am not saying that this is definitely what the authors believe, nor that it is true, but that is how their message came across to me as a reader. If true (and I would not actually be surprised if it is), this suggests that the paper might be more appropriate for an acoustics journal like JASA, rather than a solid Earth journal. Otherwise, the authors must better justify how this work advances solid Earth science.*

*It may be that the authors think that these methods could be extended to elastic media and to significantly more complex (heterogeneous) models, and they simply have not explained how. Or it may be that they think that this research closes off an avenue of research for solid Earth science which is useful to stop others from following – in effect they might decide to argue that they prove that this approach or network architecture will not be fruitful for the Earth sciences. Either is a positive step for science. One way or the other, the authors need to explain more clearly which (or which other message) we should take from their work, and why.*

*However, since this is a Discussion paper, in fact I think the best would be to rethink the paper slightly: I would begin by thinking through, and then presenting, a roadmap that might solve real modelling or inversion problems in the solid Earth sciences using models of more realistic complexity. Then show how this paper fits into that – either taking a first step in a possible direction to achieve it, or testing an avenue that while successful, turns out not to be practical for the real Earth. Either way, in the Discussion section they can explain how this work has advanced our state of knowledge about the overall strategy, and how we should move forward in future.*

Author response:

We can see how our message comes across as though this is a toy problem that cannot be extended to elastic media and that the paper does not consider enough how it advances Solid Earth science.

The message we want to convey is that we are taking the first steps in understanding whether deep learning can aid seismic simulation tasks relevant to the Solid Earth sciences, and in that sense proposing a new research direction.

Whilst we only present methods for "toy" 2D acoustic simulation, and whilst we believe there are challenges to extending our methods to 3D, more heterogeneous media and elastic simulations tasks relevant to the Solid Earth, based on our first steps we believe further research is needed and the avenue could eventually yield practical and useful tools for Solid Earth science. We believe this ought to be done before any decision on whether this approach has the potential to scale up at production level for realistic Earth science applications.

We currently believe that extending our method to 3D is mostly a computational challenge, rather than requiring an entire conceptual recast to our network design. Similarly, our

network structure is readily extendable to elastic simulation; the dimensionality of the network's inputs and outputs just needs to be increased. Whilst further research is needed to understand if these extensions are possible, we believe that our main contribution is to show that deep neural networks can simulate more complex 2D models for which no solutions other than numerically discretized ones exist and not just simple 1D models, which is a positive step towards using deep neural networks in real applications.

Proposed manuscript changes:

We propose to "reframe" the manuscript such that it includes a sort of "manifesto" or roadmap proposing and investigating deep learning for aiding Solid Earth simulation tasks:
1. Add more emphasis in the introduction on practical Solid Earth seismic simulation tasks, their challenges and how deep learning and our method fits into them
2. Add more detail in the discussion section on future research directions which could extend our methods so they can eventually be used in real Solid Earth simulation tasks. We would include new subsections discussing what we see are the main challenges towards this: 1) extending to more heterogeneous models, 2) extension to elastic simulation and 3) extension to 3D simulation.

Reviewer comments:

*[Reviewer 1] • What do the authors foresee as the primary application(s) of their approach? The discussion seems to mainly envisage inversion-related use cases. Some comments on this:*
1. *The 'fast seismic inversion' approach discussed in Section 3 is essentially a variant of the 'prior sampling' approach discussed in detail by Käufl et al (2016). The key strength and weakness of this as an inversion strategy is that all samples (i.e. the training set) are generated without reference to any observed data. This enables very rapid inference once data becomes avail- able, but it means that most training samples lie far from the observed data and are largely wasted from the perspective of any one inference. The end result is that inferences are considerably less well-constrained than would be possible with posterior sampling (see Fig. 9 of Käufl et al 2016). The bottom line is that prior sampling only seems a worthwhile strategy for (a) problems where time is of the essence, e.g. earthquake early warning, or (b) problems where the 'same' inference task needs to be solved many thousands of times with different data vectors.*
2. *Using the learned model in Monte Carlo simulations seems superficially attractive, but comes with significant caveats. Fundamentally the inference remains entirely based on the information contained within the training dataset, and so all the limitations of prior sampling remain. The random walk would need to be constrained to only generate models compatible with the training data, if results are to be meaningful. Perhaps it would be possible to progressively retrain the learned simulation as the McMC proceeds, to ensure accuracy in relevant parts of the model space: this starts to move towards the Bayesian optimisation approaches discussed in (e.g.) Wang et al (2013). To play devil's advocate: if a problem is too complex to tackle using an McMC approach using physical simulations, can we really be confident that a learned model is sufficiently accurate to yield meaningful results? How big a training set is required to capture the full complexity of the physical problem?*

*Some discussion and commentary on these issues, and other potential applications, would be appreciated.*

Author response:

We agree that discussing in more detail the potential applications of our method would be valuable, as described in our previous response above. We also think that the specific issues highlighted on inversion in this comment are important to consider and we propose to expand our discussion on the pros and cons of our inversion technique to include these.

Proposed manuscript changes:

1. Add more detail in the discussion section on potential applications (as described in our response above)
2. Discuss in more detail the issues of prior sampling and limitations of the training dataset when discussing the pros and cons of our inversion technique.

Reviewer comments:

*[Reviewer 1] • The main 'selling point' of the author's approach is that it enables seismograms to be generated significantly faster than would be possible using 'traditional' forward models. However, this comes with a number of caveats that I think need to be discussed more carefully.*
   1. *(As above) the author's approach is (I suspect) only effective for models that are sufficiently similar to those in the training set. The numerical forward code does not suffer from this restriction, and can handle complexities that aren't present in the authors' setup (e.g. anisotropy, variable density). How much of a speedup could be achieved by using a numerical code that had been designed with prior knowledge of the characteristics of the authors' training set? Put another way: the speedup could be made to seem even more impressive by using a code designed for a vastly more complex setting (e.g. SPECFEM3D) to build the training set. How fair is the comparison that is being presented?*
   2. *The headline speed comparisons ignore the costs of building a training set and then training the neural networks, which are significant. How many simulations does a user need to envisage performing before the author's approach becomes cost-effective overall? I think this is going to be a rather large number. Again, some discussion of the pros and cons of the author's approach would seem desirable.*

*[Reviewer 1] • In general, we can make numerical simulations faster by introducing physical approximations. In such cases we typically have some intuition for how those approximations will impact upon results. Learned models offer a speedup without explicit physical approximations, but come with uncertainties that are difficult to quantify rigorously, and which may vary considerably depending on the particular set of inputs chosen. Would the authors like to comment on the pros and cons of the two different strategies for reducing computational costs?*

*[Reviewer 2] I also agree with most (but not all) of the comments from Reviewer 1 (Andrew Valentine), in particular his comment that a fair comparison for the layered-medium example would be between neural networks and other modelling methods that intrinsically assume a 1D Earth structure rather than full finite difference methods.*

Author response:

For the first case where we consider simple layered 1D Earth models we agree that a fairer speed comparison would be against existing numerical methods which intrinsically assume 1D layers. For the 2D faulted Earth models we consider, to the best of our knowledge FD methods are the most efficient tools for these types of models, and therefore we believe this is a fair comparison.

We agree that there is more nuance to the comparison than just speed; for example, a discussion on where our approach break downs and the cost of training dataset generation and training would be useful.

Proposed manuscript changes:

1. For the case of 1D layered Earth models compare our approach (our WaveNet model) against the speed of an existing numerical method which intrinsically assumes 1D layers.
2. Include a discussion of where our approach breaks down and the relative cost of generating the training data and training our models.

Reviewer comments:

*[Reviewer 2] 4. The paper appears to have committed the equivalent of an 'inverse crime'. If I understand correctly, the authors have trained networks in the forward and inverse directions using models with a certain parametrisation, and have tested the networks on models of exactly the same parametrization (Reviewer 1 touched on this too). While this would be reasonable if the models were themselves reasonably realistic, in a practical field like Geophysics I think it is necessary to test the networks on examples that lie outside of the range of the training set – not only using different models from those in the training set, but models that are not within the span of the algorithm used to generate the training set (as in the real Earth).*

*For example, (a) Earp and Curtis (2019, arXiv – https://arxiv.org/abs/1907.00541 ) and (b) Earp et al. (2019, arXiv – https://arxiv.org/abs/1908.09588 ) perform (a) probabilistic travel time tomographic imaging, and (b) probabilistic surface wave inversion for averaged shear wave structures with depth, using deep neural networks. The test examples using in both cases are created using a finer parametrization than was used in the network training set – thus the actually structure of the (synthetically) 'true' Earth is not attainable by the networks; nevertheless, they can be used as a useful check of whether in such cases (as in the real Earth) the networks behave sensibly – giving results that are spatial averages in some sense of the 'true' structure. To be clear I do not think that the above references are perfect in this regard and could certainly be improved (e.g., could use even more complex models for tests); nevertheless, the authors could usefully think about such tests for their work as it would strengthen the conclusions.*

*[Reviewer 1] • In any ML-based approach, the training data is central to the applicability of the method. The author's trained network appears effective for simulating waveforms in models that are generated using the same criteria as were used to make the training set. However, I suspect performance will be significantly worse*

*for models that have significantly different character. This is something that deserves more discussion than it receives, perhaps with some examples. A particular concern in practical settings may be how an end-user can assess whether their input model is 'sufficiently close' to the training set.*

Author response:

Whilst we were careful not to test the performance of the network using the same examples which were used to train the network, we only used examples drawn from the same distribution as the training distribution to test our networks. Because the training distribution only contains simple models we agree with the reviewers that this does not inform us on the performance of the network for models outside of the training set, and furthermore our current networks are likely to perform worse for such models.

For many seismological applications of forward and inverse modelling, we believe that the Earth models used are typically within a known range of parameters and therefore a training set could be constructed which appropriately spans the expected models, however we believe more research is needed in this area. For instance, dozens of published tomographic models could be used to define a base model range for future tomography modelling where thousands or millions of similar model simulations are needed.

Proposed manuscript changes:

1. We propose to test both our models on more realistic Earth models outside of the span of our training distribution and show the degradation observed. We also propose to suggest ways (such as nearest neighbour analysis) which could help a user determine how close an input model is to the training distribution, and potential future research ideas which could quantify the prediction uncertainty.
2. We will add more discussion on potential applications where we think this limitation is and is not permissible.

Reviewer comments:

*[Reviewer 2] 2. The introduction is interesting and reviews some of the appropriate material, but is very sparsely justified, and as also stated by Reviewer 1, it does not include many key references. In my view, every sentence of a scientific work must either be a logical deduction from previous text, must have been deduced/proven in another paper, or may be an argument based on the material in another paper; in that latter two cases that paper needs to be discussed and cited. In this paper, the Introduction cites very few references and therefore contains unjustified (in the sense of, 'not justified') statements. Examples include:*

*One cannot write a paper on using neural networks to perform full waveform inversion (FWI) without citing Roeth and Tarantola (1994 – J. Geophys. Research). How does the FWI part of this paper improve on their work? That is not at all clear. There are many other papers using neural networks for imaging in Geophysics using waveforms or other types of data; you need to read and cite them, and describe how this work advances the field relative to those works. Currently the latter is not clear. The authors make the case for using neural networks for real-time applications – again first steps in this direction have already been taken (see Cao et al., 2019 – Geophysics, for example) and should be discussed.*

*[Reviewer 1] • Referencing, especially in the introduction, seems rather haphazard. If citations are to be given for broad, well-established topics such as the utility of seismic simulations in reservoir characterisation, I would expect these to be to major review papers or to 'classics': these are going to be most useful for a reader who is unfamiliar with the field. Without intending any criticism of the cited works, this does not really seem to be the case at present. Moreover, the authors' survey of the history of machine learning in geophysics is very short-sighted, ignoring anything more than a couple of years old. There are neural network papers in the geophysical literature from the late 1970s onwards, and it would be nice to see some acknowledgement of this body of work. Valentine & Trampert (2012) is probably the first instance of 'deep learning' in seismology, though the term had not been invented at that point (and we did not have the benefit of modern computational frameworks).*

Author response:

We agree that the referencing in the introduction is sparse and that we only review examples of applications of deep learning to geophysics in the last couple of years, and that the paper would be stronger with more discussion of relevant work.

Proposed manuscript changes:

1. Ensure citations on broad, well-established topics in the introduction are major review papers or "classics".
2. Add and discuss more references where they are sparse, for example Roeth and Tarantola (1994 – J. Geophys. Research) when discussing FWI, and Cao et al., 2019 – Geophysics for real time simulation.
3. Include a fuller review of the applications of deep learning in geophysics, and include earlier examples, such as Valentine & Trampert (2012) and Devilee et al., (1999 – J. Geophys. Res).

Reviewer comments:

*[Reviewer 2] 3. The authors promote the fact that they use 'deep learning', and their application certainly fits into that category. However, they must at least discuss why this is a positive feature of the method, and cite previous Geophysical applications of deep learning to support that discussion. Deep learning is usually defined to be the use of 4 or more layers within a neural network. While I agree with Reviewer 1 that his previous work (Valentine and Trampert) was an example of deep learning, the first that I know of in Geophysics was in fact Devilee et al., (1999 – J. Geophys. Res) – which came from the same university.*

*In my view there is therefore nothing new about the concept of deep learning: we were using it in Geophysics in the '90's. What has changed is the extent to which depth can be used to impose useful structure on networks (as the authors themselves have done in this manuscript – their Figure 9, and also in the paper cited by Reviewer 1); also the number of parameters that can now be used (the width of each layer) has increased hugely. In fact the number of parameters in the authors' application is relatively modest compared to some in machine learning literature, but is certainly comparable to other recent studies in Geophysics; the structure that the authors impose is both sensible and clearly useful in order to help to obtain stable results. These things should be discussed.*

Author response:

We agree that deep learning is not a new technique in Geophysics and that two of the enabling recent advancements in this field are the ability to train deeper models with many more parameters. We agree the manuscript does not explicitly make this distinction and we propose to make this clearer.

Proposed manuscript changes:

1. Explicitly acknowledge that deep learning concepts are not new in geophysics and have been used in the past, and explain that, among other factors, such as the availability of more powerful hardware, advancements in training deep models with more parameters have enabled this work.

---

## Author Response (AR2)

Author response to referee comments (revised submission)

Deep learning for fast simulation of seismic waves in complex media

Introduction:
 This document contains our responses and proposed manuscript changes resulting from the two reviews of our revised "Deep learning for fast simulation of seismic waves in complex media" discussion paper.

We would like to thank the reviewers (Andrew Curtis and Andrew Valentine) for reviewing our revised manuscript and our response to their comments are below.

Ben Moseley, Tarje Nissen-Meyer & Andrew Markham

Report 1

This manuscript has evolved significantly since its original submission, and I thank the authors for their careful consideration of the reviews. A few comments:

- The authors frame their manuscript as 'a manifesto or roadmap for deep learning'. I think this is a good line to take, and their work certainly makes a convincing case for further exploration. However, it should be noted that the manuscript really only explores one or two possible ways of framing seismic problems using deep learning. On occasion the authors' language creeps beyond this, towards a claim that they have comprehensively characterised the space of possibilities.

Authors: We agree there are likely to be many alternative ways to frame this task using deep learning, for example by using other network types. We propose to add a sentence in the final summary (Section 4.5) acknowledging this explicitly and stating that we only consider a few types of networks (WaveNet and conditional autoencoders).

- I think the comparisons between the wavenet-derived simulations and 2D ray tracing are very interesting. If I understand Fig. A1 correctly, the authors' approach consistently out-performs the 'physical' method in terms of accuracy/fidelity to the training data. This could perhaps be discussed in more detail in the main text. Looking at Fig. 5, it appears that the ray-tracing introduces a slight phase shift relative to the reference waveforms. Is this the main source of discrepancy between the two? What do the worst-performing wavenet simulations look like? Does wavenet always 'more-or-less' work, or does it sometimes fail egregiously?

Authors: The WaveNet does match the FD simulations better than the 2D ray tracing across the test set used in Figure A1. Small phase shifts and small differences in the amplitudes at large offsets are the main source of discrepancy between the ray tracing and FD modelling. The improved performance of the WaveNet is perhaps to be expected because it is trained to directly match the FD simulation, instead of the ray tracing. The WaveNet is consistent in

its predictions across the test set, but performs poorer when tested outside of its training distribution (Figure 6). We propose to add these observations to Section 2.6.

- To what extent have the authors explored different deep learning formulations? Does using a smaller NN cause gradual deterioration or total failure of the approach?

Authors: We explored multiple different network architectures and loss functions for the conditional autoencoder in Figure A2; a smaller NNs ("thin"and "shallow" cases in figure) causes gradual deterioration, whilst changing the gain value from 2.5 to 5 or 0 in the loss function causes total failure. We also explored different network architectures for the WaveNet (Figure A1; without exponential dilations causes total failure, whilst removing its causality constraint slightly reduces performance). We comment on these differences in Section 2.6 and 3.3 and believe they are covered well in the manuscript.

- The section on 'inverse wavenet' (using deep learning to map waveforms into models) has been scaled back significantly as a result of the earlier reviews. I wonder if it might be appropriate to move all discussion of this (i.e. formulation, training, results, discussion) into a single place within the manuscript, presumably within Section 4. I think this might better reflect the speculative tone of this part.

Authors: We agree this part is more speculative and moving the section is appropriate; we scaled it back to focus the paper towards simulation rather than inversion. We propose to merge Section 2.6 and its results in Section 2.7 into Section 4.4.

Finally, as a more general perspective: I would argue that the utility (or otherwise) of 'learned' approaches is highly context-dependent. In general, machine-learning--based methods offer computational costs that scale very differently from 'conventional' approaches, but with less favourable (or at least, less clearly-understood) accuracy guarantees. I think this probably means that further development of 'learned' methods needs to be targeted towards specific identified applications: the cost/benefit analysis for (say) applications in earthquake early warning may be very different from those in (say) global tomography. Optimising and validating any learned strategy ought to take this into account, and I am not sure that a one-size-fits-all approach is realistic.

Authors: We propose to highlight more clearly in Section 4.5 that deep learning has different computational costs and less well understood accuracy guarantees than conventional approaches and that these should be considered for each application.

Report 2

I think the authors have done a good job of addressing the comments from both myself and reviewer 1. The manuscript is much improved and I recommend publication with one minor addition:

When discussing the extension to 3D, the authors state: "However, this approach is likely to be practically challenging because increasing the dimensionality would increase the number of weights and likely the training time."

I think they should mention a little more explicitly/clearly that there are three compounding reasons that costs increase when moving to 3D, of which I think the increased number of weights is probably the least of our worries...:

1. the number of weights in the network increases significantly.
2. the cost of forward modelling the training examples increases hugely.
3. the "curse of dimensionality" needs to be mentioned explicitly as this is what implies that we require exponentially many more samples and hence modelling runs in order to explore parameter space.

Together these three make the extension to 3D incredibly challenging, and the worst of the lot is the Curse: it doesn't matter how fast your modelling method is, the curse can still make a problem impossible with only a few tens of parameters.

Authors: We believe the first two reasons are important factors to overcome when moving to 3D for simulation purposes, and propose to list them more explicitly in Section 4.2. We believe that the curse of dimensionality, as interpreted in the setting of an inverse, sampling, or general probabilistic inference problem, does not apply in the same sense to our forward simulation networks because we are carrying out forward simulation rather than "exploring a parameter space". Once any forward solution (deep networks, FD, other) exists, the curse of dimensionality certainly dictates the number of simulations needed to explore parameter space comprehensively and we fully agree that an MCMC-style posterior sampling is out of reach for a finely gridded 3D space, but that is beyond the scope of this paper which merely suggests that NN can lead to a speedup of the forward problem. We believe that the "manifestation" of the curse of dimensionality for the forward problem is that, for a given number of training examples, it is only possible to cover a small subset of the input space of possible velocity models in the network's training distribution, which restricts its generalisation ability. We believe we cover this in Section 4.3 when discussing generalisation issues by mentioning that trying to simulate all possible Earth models would likely become computationally intractable. We think that moving all of the inversion results to Section 4.4 (see comments above) makes it clearer that Section 4.2 only considers the difficulty of extending simulation networks to 3D, and we added a sentence to Section 4.4 acknowledging the curse of dimensionality for inverse problems.

Otherwise, very nice manuscript.

Authors: in addition to the comments above, we made some minor grammatical changes throughout the manuscript and added a sentence emphasising the motivation for using our approach at the start of Section 2.

[revised manuscript text omitted]

---

## Author Response (AR3)

Author final edits of manuscript

Deep learning for fast simulation of seismic waves in complex media

This document describes the final edits made to our manuscript ready for publication.

Ben Moseley, Tarje Nissen-Meyer & Andrew Markham

Final edits made:

We changed the code availability statement to include the link to the github repository where all the code to reproduce our results will be made available (on the publication of the paper).